# NetHack is Hard to Hack

**Ulyana Piterbarg**[*]
NYU

**Lerrel Pinto**
NYU

**Rob Fergus**
NYU

## Abstract

Neural policy learning methods have achieved remarkable results in various control problems, ranging from Atari games to simulated locomotion. However, these methods struggle in long-horizon tasks, especially in open-ended environments with multi-modal observations, such as the popular dungeon-crawler game, NetHack. Intriguingly, the NeurIPS 2021 NetHack Challenge revealed that symbolic agents outperformed neural approaches by over four times in median game score. In this paper, we delve into the reasons behind this performance gap and present an extensive study on neural policy learning for NetHack. To conduct this study, we analyze the winning symbolic agent, extending its codebase to track internal strategy selection in order to generate one of the largest available demonstration datasets. Utilizing this dataset, we examine (i) the advantages of an action hierarchy; (ii) enhancements in neural architecture; and (iii) the integration of reinforcement learning with imitation learning. Our investigations produce a state-of-the-art neural agent that surpasses previous fully neural policies by 127% in offline settings and 25% in online settings on median game score. However, we also demonstrate that mere scaling is insufficient to bridge the performance gap with the best symbolic models or even the top human players.

## 1 Introduction

Reinforcement Learning (RL) combined with deep neural policies has achieved impressive results in control problems, such as short-horizon simulated locomotion tasks [55, 7]. However, these methods struggle in long-horizon problem domains, such as NetHack [34], a highly challenging grid-world game. NetHack poses difficulties due to its vast state and action space, multi-modal observation space (including vision and language), procedurally-generated randomness, diverse strategies, and deferred rewards. These challenges are evident in the recent NetHack Challenge [23], where agents based on hand-crafted symbolic rules outperform purely neural approaches (see Figure 1), despite the latter having access to high-quality human demonstration data [24] and utilizing large-scale models.

We propose three reasons for the poor performance of large-scale neural policies compared to symbolic strategies. First, symbolic strategies implement hierarchical control schemes, which are generally absent in neural policies used for NetHack. Second, symbolic models use hand-crafted parsers for multi-modal observations, suggesting that larger networks could enhance representations extracted from complex observations. Third, symbolic strategies incorporate error correction mechanisms, which could be crucial for improving neural policies if integrated with RL based error correction.

In this work, we conduct a comprehensive study of NetHack and examine various learning mechanisms to enhance the performance of neural models. We bypass traditional RL obstacles, such as sparse rewards or exploration challenges, by focusing on imitation learning. However, we find that existing datasets lack crucial information, such as hierarchical labels and symbolic planning traces. To address this, we augment the codebase of `AutoAscend`, the top-performing symbolic agent in the 2021 NetHack Challenge, and extract hierarchical labels tracking the agent's internal strategy selection in order to construct a large-scale dataset containing $10^9$ actions.

---

[*]Correspondence to `up2021@cims.nyu.edu`.

37th Conference on Neural Information Processing Systems (NeurIPS 2023).

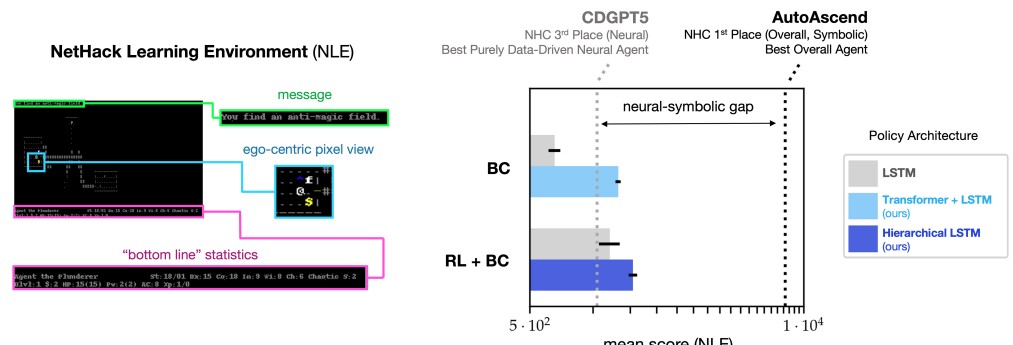

Figure 1: *Left:* The per-step observation for NetHack agents consists of an ego-centric pixel image (blue) and two text fields containing in-game messages (green) and statistics (pink). *Right:* Selected results from the NeurIPS 2021 NetHack Challenge (NHC) [23] showing game score on a log-scale. Error bars reflect standard error. Neural baseline models (grey) trained with BC perform poorly, but are somewhat improved when fine-tuned with RL. We find that the introduction of hierarchy and changes in model architecture yield significant improvements (light/dark blue), resulting in state-of-the-art performance for a neural model. However all neural approaches are significantly worse than `AutoAscend`, a hand-crafted symbolic policy. Our paper explores this performance gap.

Using this dataset, we train a range of deep neural policies and investigate: (a) the advantages of hierarchy; (b) model architecture and capacity; and (c) fine-tuning with reinforcement learning. Our main findings are as follows:

- Hierarchical behavioral cloning (HBC) significantly outperforms BC and baseline methods, provided that the model has adequate capacity.
- Large transformer models exhibit considerable improvements over baselines and other architectures, such as LSTMs. However, the power-law's shallow slope indicates that data scaling alone will not suffice to solve the game.
- Online fine-tuning with RL further enhances performance, with hierarchy proving beneficial for exploration.
- The combined effects of hierarchy, scale, and RL lead to state-of-the-art performance, narrowing the gap with `AutoAscend` but not eliminating it.

We validate the statistical significance of our findings with low-sample hypothesis testing (see Appendix H). Additionally, we open-source our code, models, and the `HiHack` repository[2], which includes (i) our $10^9$ dataset of hierarchical labels obtained from `AutoAscend` and (ii) the augmented `AutoAscend` and `NLE` code employed for hierarchical data generation, encouraging development.

Though we base our experiments in NetHack, we take measures to preserve the generality of insights yielded by our investigations of neural policy learning. Specifically, we intentionally forgo the addition of any environment-specific constraints in the architectural design or training setup of all models explored in this paper. This is in contrast to NetHack agents RAPH and KakaoBrain, the first and second place winners of the neural track of the NHC Competition, respectively, which incorporate augmentations such as hand-engineered action spaces, role-specific training, and hard-coded symbolic sub-routines [23]. While this choice prevents us from achieving absolute state-of-the-art performance in NLE in this paper, we believe it to be crucial in preserving the general applicability of our insights to neural policy learning for general open-ended, long-horizon environments.

## 2 Related Work

Our work builds upon previous studies in the NetHack environment, imitation learning, hierarchical learning, and the use of transformers as policies. In this section, we briefly discuss the most relevant works.

---

[2]Code is available at `https://github.com/upiterbarg/hihack`.

**NetHack**   Following the introduction of the NetHack Learning Environment (NLE) [34], the NetHack Challenge (NHC) competition [23] enabled comparisons between a range of different agents. The best performing symbolic and fully data-driven neural agents were `AutoAscend` and `Chaotic-Dwarven-GPT-5` (CDGPT5), respectively, and we base our investigations on them, as well as on the NetHack Learning Dataset [24].

Several notable works make use of the NetHack environment. Zhong et al. [61] show how a dynamics model can be learned from the Nethack text messages and leveraged to improve performance; on account of their utility, we also encode the in-game message but instead use a model-free policy to pick actions. Bruce et al. [9] show how a monotonic progress function in NetHack can be learned from human play data and then combined with a reinforcement learning (RL) reward to solve long-range tasks in the game. This represents a complementary way of employing NetHack demonstration data without direct action imitation.

**Imitation Learning**   Pomerleau [47] demonstrated the potential of driving an autonomous vehicle using offline data and a neural network, which has since become an ongoing research topic for scalable behavior learning [3, 6, 50]. Formally, behavior learning is a function approximation problem, where the goal is to model an underlying expert *policy* mapping states or observations to actions, directly from data. Approaches to behavior learning can be categorized into two main classes: *offline RL* [21, 32, 33, 58, 36, 20], which focuses on learning from mixed-quality datasets with reward labels; and *imitation learning* [42, 44, 45, 26], which emphasizes learning behavior from expert datasets without reward labels. Our work primarily belongs to the latter category as it employs a behavior cloning model. Behavior cloning, a form of imitation learning, aims to model the expert's actions given the observation and is frequently used in real-world applications [60, 62, 60, 48, 18, 59].

Since behavior cloning algorithms typically address a fully supervised learning problem, they are often faster and simpler than offline RL algorithms while still yielding competitive results [20, 22]. A novel aspect of our work is the use of hierarchy in conjunction with behavioral cloning, i.e. supervision at multiple levels of abstraction, a topic which has received relatively little attention. Recent efforts to combine large language models with embodied agents use the former to issue a high-level textual "action" to the low-level motor policy. Approaches such as Abramson et al. [1] have shown the effectiveness of hierarchical BC for complex tasks in a simulated playroom settings.

**Hierarchical Policy Learning**   Hierarchical reinforcement learning (HRL) based techniques [5, 4] extend standard reinforcement learning methods in complex and long-horizon tasks via temporal abstraction across hierarchies, as demonstrated by Levy et al. [37] and Nachum et al. [40][41]. Similarly, numerous studies have concentrated on showing that primitives [17, 54, 53] can be beneficial for control. These concepts have been combined in works such as Stochastic Neural Networks by Florensa et al. [19], where skills are acquired during pretraining to tackle diverse complex tasks. Likewise, Andreas et al. [2] learn modular sub-policies for solving temporally extended tasks. However, most prior work focus on learning both levels of the hierarchy, i.e. a decomposition across primitives, skills, or sub-policies as well as the primitives, skills, or sub-policies themselves, which makes training complex. Correspondingly, the resultant approaches have had limited success on more challenging tasks and environments. Le et al. [35] explores the interaction between hierarchical learning and imitation and finds benefits, albeit in goal-conditioned settings. In contrast, our work explores the benefits of access to a fixed hierarchy chosen by the domain-expert designer of `AutoAscend`, which simplifies our study of overall learning mechanisms for NetHack.

**Transformers for RL**   The remarkable success of transformer models [57] in natural language processing [15, 8] and computer vision [16] has spurred significant interest in employing them for learning behavior and control. In this context, [11, 29] apply transformers to RL and offline RL, respectively, while [12, 14, 38] utilize them for imitation learning. Both [14, 38] primarily use transformers to summarize historical visual context, whereas [12] focuses on their long-term extrapolation capabilities. More recent work have explored the use of multi-modal transformers [27] to fit large amounts of demonstration data [49, 52, 56]. To enable transformers to encode larger context lengths, recurrent transformer models have been proposed [13, 10]. Our work draws inspiration from these use cases, employing a transformer to consolidate historical context and harness its generative abilities in conjunction with a recurrent module.

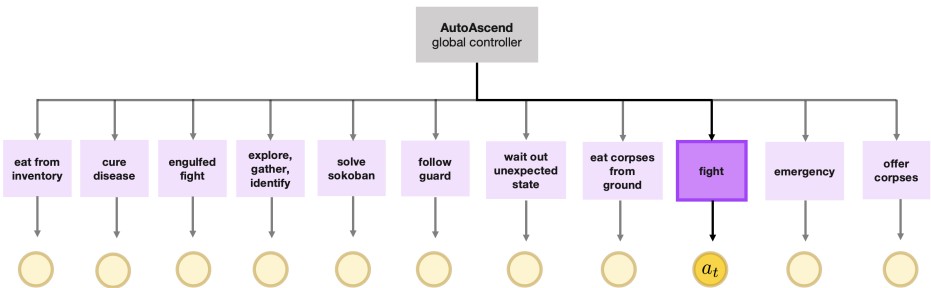

Figure 2: A diagrammatic visualization of the internal structure of `AutoAscend`. The bot is composed of eleven goal-directed, high-level *strategies*. The "global controller" underlying `AutoAscend` employs a complex predicate based control flow scheme to determine which strategy to query for an action on a per-timestep basis [23].

# 3 Data Generation: Creating the `HiHack` Dataset

## 3.1 Extending the NetHack Learning Environment

The NetHack Learning Environment (NLE) is a `gym` environment wrapping the NetHack game. Like the game itself, the action and state spaces of NLE are complex, consisting of 121 distinct actions and ten distinct observation components. The full observation space of NLE is far richer and more informed than the view afforded to human players of NetHack, who observe only the more ambiguous "text based" components of NLE observations, denoted as `tty_chars`, `tty_colors`, and `tty_cursor`. This text based view corresponds also to the default format in which both NetHack and NLE gameplay is recorded, loaded, and streamed via the C based `ttyrec` library native to the NetHack game.

The popular NetHack Learning Dataset (NLD) offers two large-scale corpuses of NetHack gameplay data, `NLD-AA`, consisting of action-labeled demonstrations from `AutoAscend`, and `NLD-NAO`, consisting of unlabeled human player data [24]. NLD adopts the convention of recording only `tty*` components of NLE observations as a basis for learning, hence benefiting from the significant speedups in data operations offered via integration with the `ttyrec` library. We adhere to this convention with the hierarchical `HiHack` Dataset introduced in this paper. Thus, in order to generate our dataset, we extend the `ttyrec` library to store hierarchical *strategy* or, equivalently, *goal* labels alongside action labels. We further integrate this extension of `ttyrec` with NLE, modifying the gym environment to accept an additional hierarchical label at each step of interaction. This input hierarchical label does not affect the underlying state of the environment, and is instead employed strictly to enable the recording of hierarchically-informed NetHack game-play to the `ttyrec` data format.

## 3.2 AutoAscend: A Hierarchical Symbolic Agent

An inspection of the fully open-source code base underlying the `AutoAscend` bot reveals the internal structure of the bot to be composed of a directed acyclic graph of explicitly defined *strategies*. The bot's underlying *global controller* switches between strategies in an imperative manner via sets of strategy-specific predicates, as visualised in Figure 2. Among these strategies are hand-engineered routines for accomplishing a broad range of goals crucial to effective survival in the game and successful descent through the NetHack dungeons. These include routines for fighting off arbitrary monsters, selecting food that is safe to eat from an agent's inventory, and efficiently exploring the dungeon while gathering and identifying valuable items, among many others. The various strategies are supported in turn by shared *sub-strategies* for accomplishing simpler "sub-goals" (see Appendix B for the full graph).

We exploit this explicit hierarchical structure in the generation of `HiHack`, extending the `AutoAscend` codebase to enable per-step logging of the strategy responsible for yielding each action executed by the bot, as supported by our modifications to the C based `ttyrec` writer library and NLE.

### 3.3 The HiHack Dataset

Our goal in generating the HiHack Dataset (`HiHack`) is to create a hierarchically-informed analogue of the large-scale `AutoAscend` demonstration corpus of NLD, `NLD-AA`. Thus, as previously described, `HiHack` is composed of demonstrations recorded in an extended version of the `ttyrec` format, consisting of sequences of `tty*` observations of the game state accompanied by `AutoAscend` action and strategy labels. `HiHack` contains a total of 3 billion recorded game transitions, reflecting more than a hundred thousand `AutoAscend` games. Each game corresponds to a unique, procedurally-generated "seed" of the NetHack environment, with `AutoAscend` playing as one of thirteen possible character "starting roles" across a unique layout of dungeons.

We verify that the high-level game statistics of `HiHack` match those of `NLD-AA` in Table 1. Indeed, we find a high degree of correspondence across mean and median episode score, total number of transitions, and total number of game turns. We attribute the very slightly diminished mean scores, game transitions, and turns associated with `HiHack` to a difference in the value of the NLE timeout parameter employed in the generation of the datasets. This parameter regulates the largest number of contiguous keypresses failing to advance the game-state that is permitted before a game is terminated. The value of the timeout parameter was set to 1000 in the generation of `HiHack`.

Table 1: A comparison of dataset statistics between `NLD-AA` [23] and our generated `HiHack` Dataset, produced by running `AutoAscend` in NLE v0.9.0 with extended `tty*` observations.

|                                 | NLD-AA        | HiHack        |
| ------------------------------- | ------------- | ------------- |
| Total Episodes                  | 109,545       | 109,907       |
| Total Transitions               | 3,481,605,009 | 3,244,729,367 |
| Mean Episode Score              | 10,105        | 8,166         |
| Median Episode Score            | 5,422         | 5,147         |
| Median Episode Game Transitions | 28,181        | 27,496        |
| Median Episode Game Turns       | 20,414        | 19,991        |
| Hierarchical Labels             | ✗             | ✓             |

## 4  Hierarchical Behavioral Cloning

Our first set of experiments leverage the hierarchical strategy labels recorded in `HiHack` for offline learning with neural policies, via *hierarchical behavior cloning* (HBC).

**Method**  Mimicking the imperative hierarchical structure of `AutoAscend`, we introduce a bilevel hierarchical decoding module over a popular NetHack neural policy architecture, namely the `ChaoticDwarvenGPT5` (`CDGPT5`) model. This model achieved 3rd place in the neural competition track of the NeurIPS 2021 NetHack Challenge when trained from scratch with RL, making it the top-performing fully data-driven neural model for NetHack [23].

The `CDGPT5` model consists of three separate encoders: a 2-D convolutional encoder for pixel-rendered visual observations of the dungeon $o_t$, a multilayer perceptron (MLP) encoder for the environment message $m_t$, and a 1-D convolutional encoder for the bottom-line agent statistics $b_t$. These three observation portions are extracted from the `tty*` NLE observations of `HiHack`. The core module of the network is an LSTM, which is employed to produce a recurrent encoding of an agent's full in-game trajectory across what may be hundreds of thousands of keypresses, both in training and at test-time. The core module may also receive a one-hot encoding of the action executed at the previous time-step $a_{t-1}$ as input.

Our hierarchically-extended version of this LSTM based policy is shown in Figure 3(left). We replace the linear decoder used to decode the LSTM hidden state into a corresponding action label in the `CDGPT5` model with a hierarchical decoder consisting of (i) a single "high level" MLP responsible for predicting the strategy label $g_t$ given the environment observation tuple $\{m_t, o_t, b_t\}$, and (ii) a set of "low level " MLPs, one for each of the discrete strategies in the `AutoAscend` hierarchy (see Figure 2), with a SoftMax output over discrete actions. The strategy prediction $g_t$ selects which of these low-level MLPs to use.

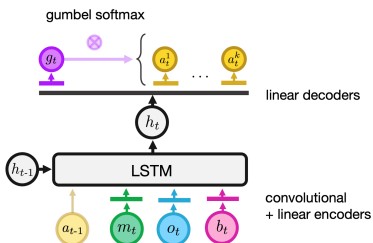
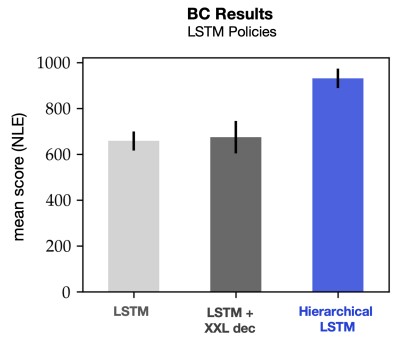

Figure 3: *Left:* Hierarchical LSTM policy, where $g_t$ is the high-level strategy prediction (purple) that is used to select over the $k$ low-level policies (yellow). Figure 1 shows the three different components of the input observation: message $m_t$, ego-centric pixel view of the game $o_t$, and "bottom line" statistics $b_t$. *Right:* Mean score for baseline LSTM model [23] (grey), our hierarchical model (blue) at the conclusion of training. The addition of hierarchical labels in decoding provides a significant performance gain, not matched by an (extra) large-decoder version of the baseline (dark grey). Error bars indicate standard error.

We employ a simple cross-entropy loss to train both the baseline non-hierarchical LSTM `CDGPT5` policy, as well as our Hierarchical LSTM policy, aggregating gradients across the bilevel decoders of the latter via the Gumbel-Softmax reparameterization trick [28].

**Training and evaluation details**    We train all policies on a single GPU for 48 hours with the full $3.2B$ HiHack Dataset. As with all experiments in the paper, a total of 6 seeds are used to randomize dataloading and neural policy parameter initialization. We employ mean and median NLE score on a batch of withheld NLE instances ($n = 1024$) as our central metrics for evaluating model performance and generalization at the conclusion of training, following the convention introduced in the NetHack Challenge competition [23]. Reported performance is aggregated over random seeds. Further details of architectures as well as training and evaluation procedures can be found in Appendices D, E, and F of the supplemental material.

**Results**    We find that the introduction of hierarchy results in a significant improvement to the test-time performance of LSTM policies trained with behavioral cloning, yielding a 40% gain over the baseline in mean NLE score as shown in Figure 3(right), and 50% improvement in median score across seeds as shown in Table 2. Additionally, to verify that this improvement in performance is indeed due to hierarchy and not simply a result of the increased parameter count of the hierarchical LSTM policy decoder, we run ablation experiments with a modified, large-decoder version of the baseline (non-hierarchical) policy architecture. The results, shown in Figure 3(right), show that increasing the size of the LSTM decoder, without the introduction of a hierarchy, does not result in performance improvements over the baseline.

## 5    Architecture and Data Scaling

Despite the benefits of introducing hierarchical labels, the performance of the Hierarchical LSTM policy trained with HBC remains significantly behind that of symbolic policy used to generate the `HiHack` demonstrations in the first place, `AutoAscend`. This finding prompts us to explore scaling: perhaps increasing the quantity of demonstration data or the model capacity may close the observed performance gap.

**Method**    To test this new hypothesis, we conduct a two-pronged investigation: (i) to explore model capacity, we develop a novel base policy architecture for NetHack that introduces a transformer module into the previous `CDGPT5` based architecture; and (ii) for data scaling, we run a second set of "scaling-law" [30] experiments that use subsets of the HiHack Dataset to quantify the relationship between dataset size and the test-time performance of BC policies.

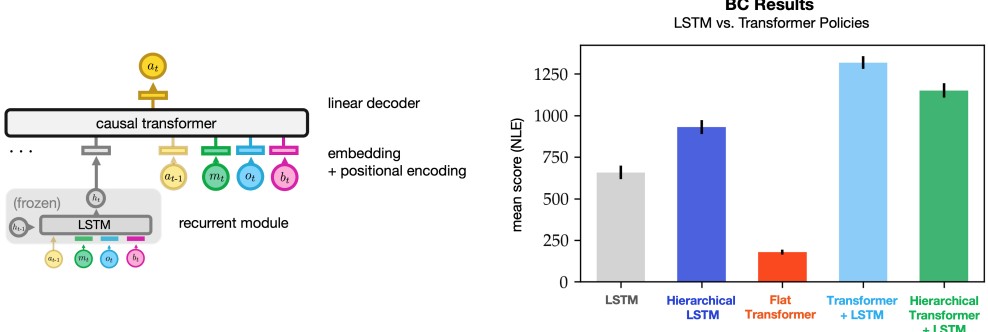

Figure 4: *Left:* Transformer based architecture (non-hierarchical version). The LSTM encoder (grey) is used to provide a long temporal context $h_t$ to the transformer. *Right:* The transformer-LSTM models (light blue and green) outperform pure LSTM models with & without hierarchy (see Section 4 and [23] respectively). Ablating the LSTM encoder component from transformer-LSTM models to yield a *flat transformer* policy architecture (orange) causes a substantial decline in policy performance. Error bars indicate standard error.

Our novel base policy architecture is visualized in Figure 4 (left). This architecture features two copies of the encoders employed in `CDGPT5`. One set is kept frozen and employed strictly to yield a recurrent encoding of the complete trajectory up to the current observation step via a pre-trained frozen LSTM, while the second is kept "unlocked" and is employed to provide embeddings of observations directly to a causal transformer, which receives a fixed context length during training.

**Training and evaluation details**  The training and evaluation procedures employed here echo those of section 4. In our data-scaling experiments, the subset of sampled `HiHack` games seen in offline training is randomized over model seeds. The causal transformer component of our transformer-LSTM models is trained with the same fixed, context-length ($c = 64$) in all experiments. Hyperparameter sweeps were employed to tune all transformer hyperparameters (see Appendix D).

**Results**  The architecture experiments in Figure 4(right) show that both variants of our combined transformer-LSTM policy architecture yield gains eclipsing those granted solely by the introduction of hierarchy in the offline learning setting.

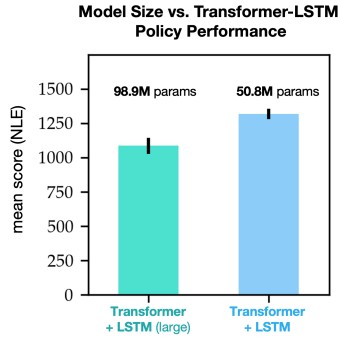

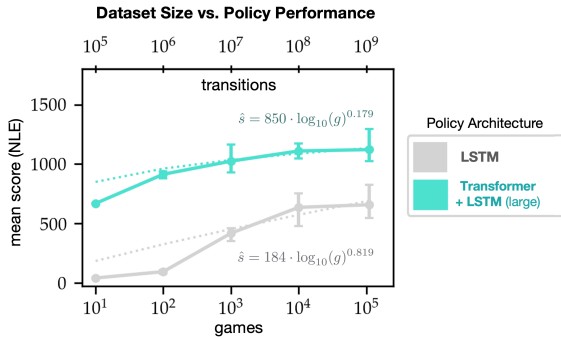

Figure 5: *Left:* Model capacity versus mean score for our transformer-LSTM model. The larger model performs worse. Error bars indicate standard error. *Right:* Dataset scaling experiments showing diminishing returns in mean NLE score $s$ achieved during training with BC as the number of training games $g$ reaches $10^5$. We employ non-linear least-squares to estimate the power law coefficients relating log-game count and mean policy score outside of the low-data regime ($g > 10^2$). Curves of best fit are displayed. Errors bars indicate mean score ranges across seeds. Collectively these two plots show that scaling of data and model size are not sufficient to close the performance gap to symbolic models.

We find the LSTM recurrent module of our transformer-LSTM models to be a crucial component of the architecture. When this module is ablated to yield a *flat transformer*, we observe a severe decline in policy performance (Figure 4(right) and Appendix G, Figure 14).

Probing further, in Figure 5(left), we compare the performance of two variants of our transformer-LSTM model, one with 3 layers (50.8M parameters) and another "scaled-up" model with 6 layers (98.9M parameters). The larger model can be seen to perform worse in evaluation than the smaller one, suggesting over-fitting and an inverse relationship between parameter count and generalization in NetHack for this model class (see also Appendix G, Figure 12). This finding indicates that scaling of model capacity alone may not be sufficient to close the neural-symbolic gap.

In Figure 5(right), we now explore the effect of training set size on mean NLE test score. We perform BC training for the LSTM baseline [23] and our largest 6 layer transformer-LSTM model (98.9M params) for 10 up to 100,000 games, subsampled from the `HiHack` Dataset. For both architectures, we observe a **sub log-linear** dependence on training set size, asymptoting at a mean score of approximately 1000. Thus, brute force scaling of the dataset alone cannot viably close the gap to symbolic methods (score of 8500).

Though our architecture and data scaling experiments are compute-time constrained, we find the test-time performance of all tested models to saturate on the given computational budget (Appendix G, Figures 11 and 12).

Table 2: Evaluating the impact of **hierarchical labels** and **architectural improvement** on the performance of policies trained both with behavioral cloning, as well as with combined behavioral cloning and asynchronchronous proximal policy optimization. All policies were trained for 48 hours on a single GPU. Metrics annotated with (†) were computed only for the top-scoring neural policy seed (out of 6) across each model class.

| | | Hierarchy | Score | | Dlvl (†) | Turns (†) | |
| --- | --- | --- | --- | --- | --- | --- | --- |
| | | | Mean | Median | Mean | Mean | Median |
| BC | LSTM [23] | ✗ | 658 ± 41 | 403 | 1.11 ± 0.01 | 5351 ± 76 | 4111 |
| BC | LSTM | ✓ | 931 ± 42 | 614 | 1.09 ± 0.01 | 6983 ± 84 | 5981 |
| BC | Transformer-LSTM | ✗ | **1318 ± 38** | **914** | **1.36 ± 0.01** | 6088 ± 75 | 5121 |
| BC | Transformer-LSTM | ✓ | 1151 ± 43 | 731 | 1.26 ± 0.01 | **7568 ± 99** | **6242** |
| APPO + BC | LSTM [23] | ✗ | 1204 ± 138 | 779 | 1.07 ± 0.01 | 8712 ± 112 | 7376 |
| APPO + BC | LSTM | ✓ | **1551 ± 73** | **972** | 1.09 ± 0.01 | **11435 ± 134** | **9849** |
| APPO + BC | Transformer-LSTM | ✗ | 1326 ± 28 | 887 | 1.25 ± 0.01 | 7924 ± 99 | 6788 |
| APPO + BC | Transformer-LSTM | ✓ | 1346 ± 16 | 894 | **1.32 ± 0.01** | 7874 ± 101 | 6769 |
| Symbolic | `AutoAscend` | ✓ | 8556 ± 187 | 4918 | 3.10 ± 0.04 | 19586 ± 171 | 19710 |

# 6 Combining Imitation with Reinforcement Learning

Given that hierarchy and scaling are insufficient to bridge the performance gap with `AutoAscend`, we now explore the impact of incorporating an online learning component to neural policy training, via RL.

**Method** In this set of experiments, we build on results from Hambro et al. [24], which suggested that in the presence of action-labeled data, behavioral cloning coupled with reinforcement learning is superior to RL training from scratch in NetHack. As in Hambro et al. [24], we employ the asynchronous `moolib` distributed-RL library to train our models with a combination of **BC and asynchronous proximal policy optimization (APPO)** [39, 46, 51]. At each time-step of training, the overall loss used to perform model parameter updates is a weighted combination of BC and RL losses, i.e. the cross-entropy loss of a batch of demonstrations from `HiHack` plus an RL loss over a batch of rollouts of the current policy in NLE. For hierarchical models, we use RL only to update the low-level strategies; that is, the strategy selection policy is trained with HBC alone and is not updated with RL.

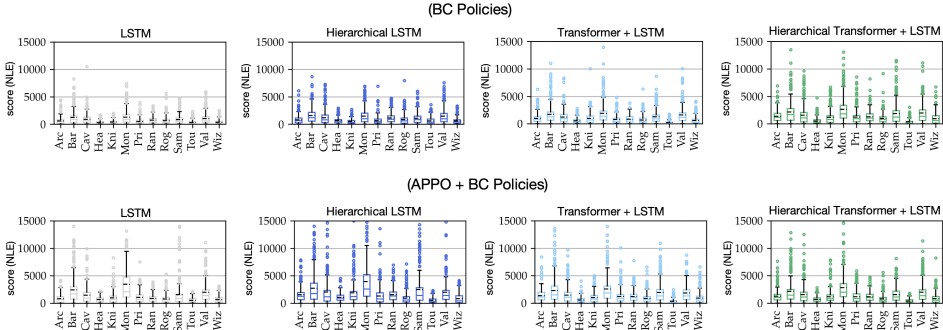

Figure 6: Aggregate NLE score breakdown versus player role. Our model refinements (hierarchy, transformer, RL fine-tuning) show gains over the LSTM based `CDGPT5` baseline [23] across all roles. As in Table 2, we employ (†) to confer that these score distributions were computed only for the top-performing neural policy seed (out of 6) across each model class.

**Training and evaluation details** Our high-level training procedure here mirrors that of our hierarchical behavioral cloning experiments: we evaluate model performance under the constraint of computation time, training all policies for exactly 48 hours on a single GPU, using 6 random seeds to randomize data loading and environment seeding only. We test the effect of warm-starting from a checkpoint pre-trained with BC or HBC alone (via the procedure delineated in section 4) for all model classes. We report APPO + BC results with warm-starting for all model classes where we find it to be beneficial in improving stability and/or evaluation policy performance.

**Results** Table 2 summarizes the performance of all our models. A number of observations can be made: (a) In the offline setting, our best performing model (non-hierarchical transformer-LSTM) outperforms the vanilla `CDGPT5` model baseline by 100% in mean NLE score and 127% in median NLE score; (b) RL fine-tuning offers a clear and significant performance boost to all models, with gains in the test-time mean NLE score associated with all model classes; (c) The overall best performing approach is APPO + BC using the hierarchical LSTM model. The mean score of 1551 represents a new state-of-the-art for neural policies in NLE, beating the performance of the vanilla `CDGPT5` model when trained with APPO + BC by 29% in mean NLE score and 25% in median NLE score; (d) The transformer-LSTM models, which are slower and more unstable to train with APPO + BC than LSTM models (see training curves in Appendix G), see a smaller margin of improvement during RL fine-tuning; (e) Other metrics commonly employed to evaluate meaningful NetHack gameplay in previous work, such as dungeon level reached and lifetime of the agent in game turns [34, 23], show a broadly similar pattern to the chief median and mean score metrics employed in the NetHack Challenge [23]; and (f) Lastly, for BC, hierarchy seems to hurt performance for the larger Transformer-RL models, though this gap is closed once APPO fine-tuning is applied. See Appendix H for low-sample hypothesis tests validating the statistical significance of these findings.

In NetHack, the player can choose from 13 distinct roles (barbarian, monk, wizard, etc.), each of which require distinctive play-styles. In NLE, starting roles are randomized, by default. Figure 6 shows a score distribution breakdown across role for different neural policy classes trained with BC and APPO + BC. In general, we observe that fine-tuning with RL improves the error-correction capability of models of all classes (as indicated by positive shifts in NLE score distributions) over their purely offline counterparts.

# 7 Conclusion and Discussion

In this work, we have developed a new technique for training NetHack agents that improves upon prior state-of-the-art neural models by 127% in offline settings and 25% in online settings, as evaluated by median NLE score. We achieve this by first creating a new dataset, the HiHack Dataset (`HiHack`), by accessing the best symbolic agent for NetHack. This dataset, combined with new architectures, allows us to build the strongest purely data-driven agent for NetHack as of the writing of this paper.

More importantly, we analyze several directions to improve performance, including the importance of hierarchy, the role of large transformer models, and the boosts that RL could provide. Our findings are multifaceted and provide valuable insights for future progress in training neural agents in open-ended environments and potentially bridging the gap to symbolic methods, as well as to the very best human players.

- Hierarchy can improve underfitting models. Prior LSTM based models severely underfit on the `HiHack` Dataset (see training curves in Appendix G). The addition of hierarchy improves such models, whether trained offline or online (Figure 3(right) and Table 2).

- Hierarchy can hurt overfitting models. Transformer based models are able to overfit on the `HiHack` Dataset (see training curves in Appendix G). Hierarchy hurts this class of models when trained offline at test-time (Figure 4(right) and Table 2).

- Reinforcement learning provides larger improvements on light, underfitting models. We obtain little and no improvement with RL fine-tuning on our hierarchical transformer-LSTM and non-hierarchical transformer-LSTM models, respectively. However, the underfit LSTM models enjoy significant gains with RL, ultimately outperforming transformer based models under a compute-time constraint (Table 2).

- Scale alone is not enough. Our studies on increasing both model and dataset size show sub-log-linear scaling laws (Figure 5). The shallow slope of the data scaling laws we observe suggests that matching the performance of `AutoAscend` with imitation learning (via games played by bot) will require more than just scaling up demonstration count.

Possible avenues for future exploration include: (a) alternate methods for increasing the transformer context length to give the agent a more effective long-term memory; (b) explicitly addressing the multi-modal nature of the demonstration data (i.e. different trajectories can lead to the same reward), which is a potential confounder for BC methods. Some forms of distributional BC (e.g. GAIL [26], BeT [52]) might help alleviate this issue.

Finally, we hope that the hierarchical dataset that we created in this paper may be of value to the community in exploring the importance of hierarchy and goal-conditioning in long-horizon, open-ended environments.

## 8  Acknowledgements

We would like to thank Alexander N. Wang, David Brandfonbrener, Nikhil Bhattasali, and Ben Evans for helpful discussions and feedback. Ulyana Piterbarg is supported by the National Science Foundation Graduate Research Fellowship Program (NSF GRFP). This work was also supported by grants from Honda and Meta as well as ONR awards N00014-21-1-2758 and N00014-22-1-2773.

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

## A   NetHack Learning Environment

Both NetHack and the NetHack Learning Environment (NLE) feature a complex and rich observation space. The full observation space of NLE consists of many distinct (but redundant) components: `glyphs`, `chars`, `colors`, `specials`, `blstats`, `message`, `inv_glyphs`, `inv_strs`, `inv_oclasses`, `screen_descriptions`, `tty_chars`, `tty_colors`, and `tty_cursor` [34].

The HiHack Dataset (`HiHack`), as well as all RL experiments in NLE conducted in this paper, consist of and rely solely upon the `tty*` view of the game.

## B   Details on AutoAscend

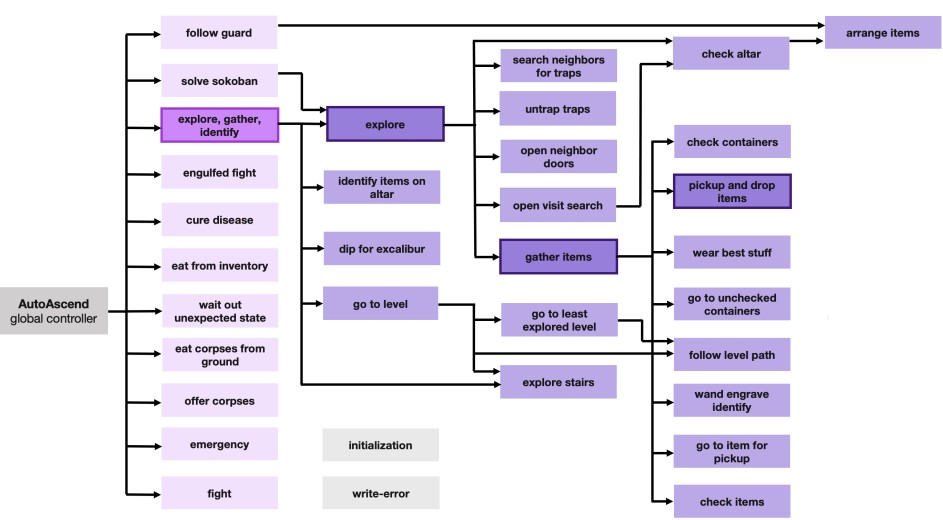

Figure 7: Full control flow structure of `AutoAscend`, separated across explicit *strategy* and *sub-strategy* routines. There are 13 possible "strategy" or hierarchical labels in `HiHack`, 11 representing the explicit *strategies* employed by the bot (in magenta) and two additional labels to handle extra-hierarchical behavior (in light-grey).

We include a comprehensive visualization of the full internal structure of `AutoAscend` in Figure 7. As indicated in the summary visualization of high-level `AutoAscend` strategies provided in Figure 2 of the main paper, the bot features 11 explicit, hard-coded *strategy* routines. These interface with other low-level *sub-strategy* routines, some which are re-used by multiple strategies or even multiple sub-strategies. One example of such a subroutine is the "arrange items" sub-strategy, which is called both by "follow guard" and by "check altar," which is itself a subroutine of the "explore" sub-strategy. When factorized across strategies and sub-strategies, the full structure of `AutoAscend` is a directed acyclic graph (DAG) with a maximal depth of 5 from the "root," i.e. the `AutoAscend` *global controller* "node" indicated in dark gray above, which re-directs global behavioral flow across strategies via a predicate-matching scheme.

The HiHack Dataset includes a hierarchical strategy *label* for each timestep of `AutoAscend` interaction. As a result, alongside the 11 explicit strategies of the bot, there are two additional labels present in the dataset, which account for extra-hierarchical behavior in `ttyrec` game records yielded by the augmented `ttyrec` writer employed for `HiHack` generation and loading. These are visualized in light gray in Figure 7. The first of these corresponds to the hard-coded initialization routine employed by `AutoAscend`, effectively serving as a twelfth (albeit implicit) strategy, while the second covers `ttyrec` timestep records with missing strategy values. Missing strategy values may reflect `ttyrec` writer errors, or advancement of the underlying NetHack state by NLE rather than by agent, which occurs e.g., during NLE's timeout based termination of games [34]. Empirically, "write-error" strategy labels occur with very low frequency in `HiHack`, representing less than $\approx 0.05\%$ of all transitions.

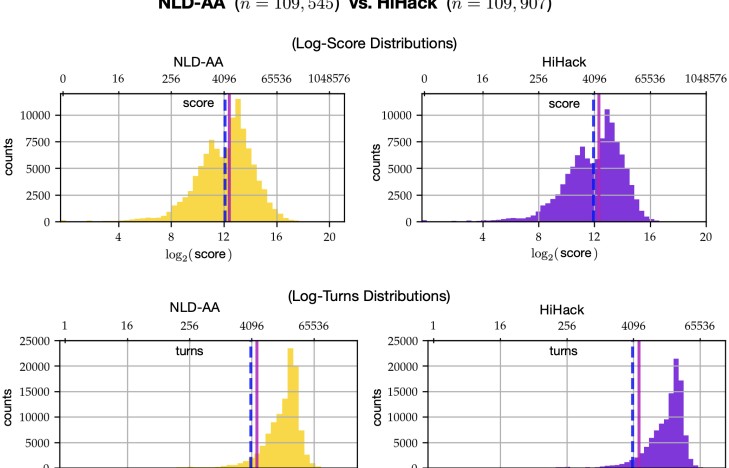

Figure 8: Distributional comparisons of basic `AutoAscend` game statistics in `NLD-AA` vs `HiHack`. Median and mean values (as reported in Table 1) are indicated respectively by vertical dashed-blue and solid-pink lines in each figure. *Top*: Log-score vs game counts in `NLD-AA` and `HiHack`. *Bottom*: Log-turns vs game counts in `NLD-AA` and `HiHack`.

## C   Details on HiHack

**Generation**   All games in the HiHack dataset were recorded by running an augmented version of `AutoAscend` in NLE v `0.9.0`. This augmented version of the `AutoAscend` source features the introduction of only a dozen extra lines of code that enable step-wise logging of the strategy trace behind each action executed by the bot. This strategy trace is recorded directly to game `ttyrecs` at each timestep via the addition of an extra channel to the C based `ttyrec`-writer in the NetHack source code. Each game was generated via a unique NLE environment seed.

**Game Statistics**   The comparison of the full log-score and log-turns distributions across `NLD-AA` and `HiHack` made in Figure 8 further supports the claim of high correspondence between the datasets made in Section 3.3. Figure 9 shows the distribution of strategies across a sample consisting of $\approx 10^8$ unique game transitions from `HiHack`. We observe coverage of all but the least frequent explicit strategy executed by `AutoAscend`: the "solve sokoban" routine, employed a means to gain yet more experience exclusively in highly advanced game states.

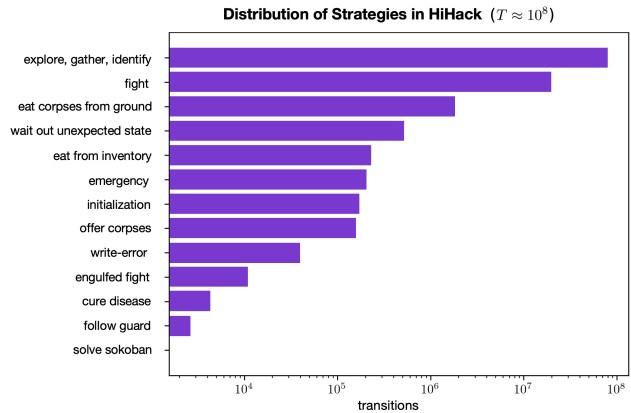

Figure 9: A visualization of the distribution of strategies across a sample of 4300 `HiHack` games, containing a total of $\approx 10^8$ transitions.

Table 3: **Training hyperparameter configurations** across all BC and APPO + BC experiments. Hyperparameters are listed in alphabetical order. We employ (‡) to indicate hyperparameters only relevant for the corresponding hierarchical policy variants. The presence of the symbol '-' in lieu of a parameter value reflects the parameter's irrelevance for offline, BC experiments. All bolded hyperparameters were tuned. After tuning was complete, precisely the same sets of hyperparameters were employed to train models across individual policy classes belonging to the LSTM and transformer-LSTM based model families explored in this paper, across all BC and APPO + BC experiments. Note that the abbreviation 'CE' denotes the cross-entropy loss function.

| | BC | | APPO + BC | |
| Hyperparameter | LSTM | transformer-LSTM | LSTM | transformer-LSTM |
| --- | --- | --- | --- | --- |
| actor batch size | - | - | 512 | 256 |
| adam beta1 | 0.9 | 0.9 | 0.9 | 0.9 |
| adam beta2 | 0.999 | 0.999 | 0.999 | 0.999 |
| adam eps | 1.00E-07 | 1.00E-07 | 1.00E-07 | 1.00E-07 |
| **adam learning rate** | 0.0001 | 0.0002 | 0.0001 | 0.0001 |
| appo clip baseline | 1 | 1 | 1 | 1 |
| appo clip policy | 0.1 | 0.1 | 0.1 | 0.1 |
| baseline cost | 1 | 1 | 1 | 1 |
| crop dim | 18 | 18 | 18 | 18 |
| **discount factor** | - | - | 0.999 | 0.999 |
| entropy cost | - | - | 0.001 | 0.001 |
| env max episode steps | - | - | 100000 | 100000 |
| env name | - | - | challenge | challenge |
| fn penalty step | - | - | constant | constant |
| **grad norm clipping** | 4 | 1 | 4 | 1 |
| inference unroll length | - | - | 1 | 1 |
| loss function | CE | CE | CE | CE |
| normalize advantages | - | - | ✓ | ✓ |
| normalize reward | - | - | ✗ | ✗ |
| num actor batches | - | - | 2 | 2 |
| num actor cpus | - | - | 10 | 10 |
| penalty step | - | - | 0 | 0 |
| penalty time | - | - | 0 | 0 |
| pixel size | 6 | 6 | 6 | 6 |
| reward clip | - | - | 10 | 10 |
| reward scale | - | - | 1 | 1 |
| **RL loss coeff** | - | - | 1 | 0.001 |
| **strategy loss coeff** (‡) | 1 | 1 | 1 | 1 |
| **supervised loss coeff** | 1 | 1 | 0.001 | 1 |
| `ttyrec` batch size | 512 | 512 | 256 | 256 |
| `ttyrec` cores | 12 | 12 | 12 | 12 |
| `ttyrec` **envpool size** | 4 | 6 | 4 | 3 |
| `ttyrec` **unroll length** | 32 | 64 | 32 | 64 |
| use prev action | ✗ | ✓ | ✓ | ✓ |
| **virtual batch size** | 512 | 1024 | 512 | 512 |

# D  Training Details

**Hyperparameters**  All relevant training hyperparameter values, across model families as well as BC vs APPO + BC experiment variants, are displayed in Table 3.

To kick-start all experiments, we employed the training hyperparameter values reported in Hambro et al. [24], used to train the `CDGPT5`, LSTM based baseline across experiments. For several hyperparameters, however, additional tuning was conducted. These hyperparameters are indicated in bold in Table 3. Tuning across these hyperparameters was performed once for the "default" representative policy class from each of the LSTM and transformer-LSTM model families for all but the Model Parameter Scaling experiments[3]. After tuning was complete, hyperparameter configurations were

---

[3]Prior to the start of these experiments, additional tuning of the "adam learning rate" and "`ttyrec` batch size" hyperparameters was conducted for the transformer-LSTM (large) policy class. However, across the

fixed across all proceeding offline and combined offline + online experiments. Specifications of hyperparameter values swept over during tuning are provided in Table 4.

All models were trained with the `Adam` optimizer [31] and a fixed learning rate. We experimented with the introduction of a learning rate schedule for transformer-LSTM models, but we found no additional improvements in policy prediction error or evaluation performance at the conclusion of training to be yielded by such a schedule.

**Random Seeds**  For each of the Hierarchical Behavioral Cloning, Model Parameter Scaling, Data Scaling, and combined Imitation and Reinforcement Learning experiments described in Sections 4, 5, and 6 of this paper, a total of 6 random seeds were run across all relevant policy classes. Randomized quantities included: policy parameter values at initialization, data loading and batching order, HiHack Dataset subsampling (in data scaling experiments only), and initial environment seeding (in APPO + BC experiments only).

**Training Infrastructure and Compute**  The RPC based `moolib` library for distributed, asynchronous machine learning was employed across all experiments [39]. All data loading and batching was parallelized. Our model training code builds heavily upon the code open-sourced by Hambro et al. [24].

Experiments were run on compute nodes on a private high-performance computing (HPC) cluster equipped either with a NVIDIA RTX-8000 or NVIDIA A100 GPU, as well as 16 CPU cores. All policies were trained for a total of 48 hours. We detected no substantial differences in training frames-per-second (FPS) rates for both offline and online experiments across compute nodes in "speed-run" tests, provided nodes were under no external load. When running experiments, we did detect some variance in total optimization steps completed under the 48-hour constrained computational budget across seeds belonging to single policy classes, which we attribute to variance in external HPC cluster load during these runs.

Table 4: **Training hyperparameter tuning sweeps**. We specify the hyperparameter values tested during tuning sweeps conducted for the "default" representatives of each model class. Full specifications of final hyperparameter values employed in experiments are included in Table 3.

|  | Sweep Range |
|---|---|
| adam learning rate | $\{0.0001, 0.0002, 0.0005, 0.01\}$ |
| discount factor | $\{0.9, 0.99, 0.999, 0.9999\}$ |
| grad norm clipping | $\{0.1, 1, 4\}$ |
| RL loss coeff | $\{0.001, 0.01, 1\}$ |
| strategy loss coeff | $\{0.001, 0.01, 1, 10\}$ |
| supervised loss coeff | $\{0.001, 0.01, 1\}$ |
| ttyrec envpool size | $\{3, 4, 6\}$ |
| ttyrec unroll length | $\{16, 32, 64, 128\}$ |
| virtual batch size | $\{128, 256, 512, 1024\}$ |

# E   Model Architectures

A description of all model components and policy architectures is given in Tables 5 and 6, separated across LSTM and transformer-LSTM model families. The `PyTorch` library was used for to specify all models, loss functions, and optimizers [43].

**Additional Transformer Specifications**  All transformer modules tested in this paper consist purely of "Transformer-Encoder" layers. Each layer is configured with 16 attention heads per attention mechanism, and layer normalization is applied prior to all attention and feed-forward operations. A dropout of 0.1 is used during training [57]. Unlike the rest of the modules we employ, which use

---

set of values tested, the same values were found to be optimal for this model configuration as for the default transformer-LSTM policy; hence, only a single set of hyperparameter values for the model family is reported here.

Table 5: **LSTM** model family architectural details. The final three columns indicate the presence (or absence) of each component across the relevant policy classes, whether trained with BC or APPO + BC.

| Class | Type | Module(s) | Hidden Dim | Layers | Activ. | Copies | Policy Class | | |
|---|---|---|---|---|---|---|---|---|---|
| | | | | | | | LSTM | LSTM + XXL dec | Hier LSTM |
| Enc | Message | MLP | 128 | 2 | ELU | - | ✓ | ✓ | ✓ |
| Enc | Blstats | Conv-1D, MLP | 128 | 4 | ELU | - | ✓ | ✓ | ✓ |
| Enc | Pixel Obs | Conv-2D, MLP | 512 | 5 | ELU | - | ✓ | ✓ | ✓ |
| Enc | Action Hist | one-hot | 128 | - | | — | ✓ | ✓ | ✓ |
| Core | - | LSTM | 512 | 1 | - | - | ✓ | ✓ | ✓ |
| Dec | Default | MLP | 512 | 1 | - | - | ✓ | ✗ | ✗ |
| Dec | XXL | MLP | 1024 | 2 | ELU | - | ✗ | ✓ | ✗ |
| Dec | Hier Strat | MLP | 128 | 1 | - | - | ✗ | ✗ | ✓ |
| Dec | Hier Action | MLP | 256 | 2 | ELU | 13 | ✗ | ✗ | ✓ |

Table 6: **transformer-LSTM** model family architectural details. As in Table 5, the final three columns indicate the presence (or absence) of each component across the relevant policy classes, whether trained with BC or APPO + BC.

| Class | Type | Module(s) | Hidden Dim | Layers | Activ. | Copies | Policy Class | | |
|---|---|---|---|---|---|---|---|---|---|
| | | | | | | | Trnsfrmr + LSTM | Trnsfrmr + LSTM (large) | Hier Trnsfrmr + LSTM |
| Enc | Message | MLP | 128 | 2 | ELU | - | ✓ | ✓ | ✓ |
| Enc | Blstats | Conv-1D, MLP | 128 | 4 | ELU | - | ✓ | ✓ | ✓ |
| Enc | Pixel Obs | Conv-2D, MLP | 512 | 5 | ELU | - | ✓ | ✓ | ✓ |
| Enc | Action Hist | one-hot | 128 | - | - | - | ✓ | ✓ | ✓ |
| Enc | Recurrent | LSTM (frozen) | 512 | 1 | - | - | ✓ | ✓ | ✓ |
| Core | Default | Trnsfrmr | 1408 | 3 | GeLU | - | ✓ | ✗ | ✓ |
| Core | Large | Trnsfrmr | 1408 | 6 | GeLU | - | ✗ | ✓ | ✗ |
| Dec | Default | MLP | 512 | 1 | - | - | ✓ | ✓ | ✗ |
| Dec | Hier Strat | MLP | 512 | 1 | - | - | ✗ | ✗ | ✓ |
| Dec | Hier Action | MLP | 512 | 2 | ELU | 13 | ✗ | ✗ | ✓ |

Exponential Linear Unit (ELU) activation functions as per the original CDGPT model architecture [23], our transformer modules employ Gaussian Error Linear Unit (GeLU) activations [25]. Doubling the number of attention heads and the hidden dimension of transformer encoder layers in addition to the layer count of transformer-LSTM models resulted only in equivalent or additional policy performance degradation over that observed for the six layer model whose specifications are reported in Table 6; thus, we include only results for the latter here.

**Context Length** Models belonging to all policy classes from the LSTM family are trained by sequentially "unrolling" batched-predictions. The length of this "unrolled" sequence is held fixed throughout training, and is specified by the value of the "`ttyrec` unroll length" hyperparameter in Table 3, also referenced as $c$ in the main body of this paper.

A fixed context length is also used to train the core transformer modules of policy classes from the transformer-LSTM family, similarly specified via the "`ttyrec` unroll length" hyperparameter. We found causally masking context in transformer attention mechanisms to be greatly beneficial towards improving the generalization capability of models. The pre-trained frozen LSTM "recurrent encoder" module of these networks provides a very cheap and simple means of dramatically extending the effective context length to cover full NetHack games (which may span hundreds of thousands of keypresses) without substantially slowing model training. As delineated in Section 5 and Appendix G, we find the recurrent module to be a crucial component of these models, with its ablation resulting in a substantial performance decline.

**Hierarchical Policy Variants**    As alluded to in Tables 5 and 6, as well as in Section 4 of the main paper, all hierarchical policy variants are equipped with two sets of decoders: one high-level *strategy decoder* trained to predict the thirteen possible strategy labels in `HiHack`; as well as thirteen low-level *action decoders*, trained to predict actions corresponding to a single `HiHack` strategy across the 121-dimensional NLE action space [34]. Our hierarchical policies thus mimic the hierarchical structure of `Autoascend`, with an action decoder corresponding to each of the eleven, *explicit* strategies executed by the symbolic bot as well as two additional action decoders corresponding to the bot's "initialization" routine (an *implicit* twelfth strategy) and "write-errors," representing missing strategy labels[4], respectively, as introduced in Figure 7. A full, diagrammatic illustration of the Hierarchical LSTM policy architecture is provided in Figure 3.

In all Hierarchical Behavioral Cloning (HBC) experiments, a BC loss was computed for the strategy decoder via ground-truth, batched `HiHack` strategy labels. A separate BC loss was computed for a single action decoder over batched `HiHack` action labels, with this action decoder "selected" in an end-to-end fashion by the strategy decoder. The action decoder "selection" procedure was executed across batches by sampling predicted strategy indices from the strategy decoder with Gumbel-Softmax re-parameterization [28], thus preserving gradient flow across the bi-level hierarchical structure of policies during training. An illustration of the full Hierarchical LSTM policy architecture is provided in Figure 3.

The strategy-specific BC loss component was re-weighted (via the "strategy loss coefficent" hyperparameter, introduced in Table 3) and recombined with low-level action decoder losses to produce a single, overall HBC loss.

We resolved the presence of `ttyrec` transitions with missing strategy values, represented via the "write-error" hierarchical label in `HiHack`, by extending the action-space of the hierarchical strategy decoder and adding an additional hierarchical action decoder copy corresponding to this class of labels. It is possible that the performance of hierarchical policy variants can be further improved by instead filtering out all transitions with this property. We leave this evaluation for future work.

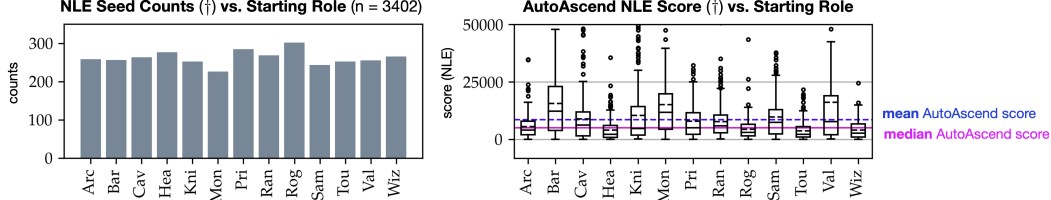

Figure 10: *Left*: The distribution of starting roles across the large-scale "in-depth evaluation." As described above, this evaluation was run for the top-performing neural policy seeds (out of 6) across model class. We observe a near-uniform distribution of possible NLE roles across random seeds. *Right*: `Autoascend` NLE Score distribution vs. starting role in the "in-depth evaluation." This figure is a companion to the visualizations of neural policy NLE scores across role in Figure 6. As in Figure 8, we indicate absolute mean and median values of `AutoAscend` NLE score in the "in-depth evaluation" with dashed-blue and solid-pink lines.

# F    Evaluation Details

For all policy classes belonging to the LSTM model family, we observe monotonic improvements in the performance of models on withheld instances of NLE as a function of training samples; as a result, we employ the *final* training checkpoint of these policies when running evaluations across policy seeds. In contrast, the generalization capabilities of models in the transformer-LSTM family do not monotonically improve as a function of training samples, which we interpret as an indicator of overfitting. Thus, evaluations are conducted only for the "best" checkpoints corresponding to each policy seed, as evaluated on the basis of the rolling NLE score proxy metric. An in-depth description of this metric, as well as experiment training curves supporting the claims of over- and underfitting across model classes, can be found in Appendix G.

---

[4]Please refer to our earlier discussion in Appendix B for more details.

Two classes of evaluations are conducted in this paper for such checkpoints: a "standard evaluation" of policy NLE score across randomly sampled and withheld instances of environment, and an "in-depth evaluation," recording all metrics of game-play and employing precisely the same set of seeded environment instances to evaluate all policies.

The former policy evaluation procedure mirrors the one conducted during the NeurIPS 2021 NetHack Challenge Competition [23]. This is the procedure we employ to compute the mean and median NLE scores associated with policy seeds for all experiments in this paper as well as to compute the estimates of `AutoAscend` mean and median NLE score in Table 2, producing our core results.

The latter policy evaluation procedure yields a suite of more fine-grained metrics for informed and "human-like" game-play in NetHack, such as maximal dungeon level reached and the total life-time of the agent. We run this evaluation procedure for each of the best-performing[5] seeds from each neural policy class, as well as for `AutoAscend`. Metrics computed with this procedure are denoted via the (†) symbol throughout the paper.

**Standard Evaluation**   Policies are evaluated on a randomly seeded batch of 1024 (withheld) NLE games. Only the final NLE scores at the end of game-play are recorded.

**In-Depth Evaluation**   Policies are evaluated across precisely the same seeded batch of 3402 (withheld) NLE games, i.e. all agents play precisely the same set of starting roles across the same NLE dungeon configurations, none of which are covered in `HiHack`. All games are recorded to the `ttyrec` data format, and can be streamed *post-facto*.

A visualization of the distribution of starting roles covered in this evaluation, as well as the corresponding `AutoAscend` score distributions (factorized by role across game instances), are shown in Figure 10.

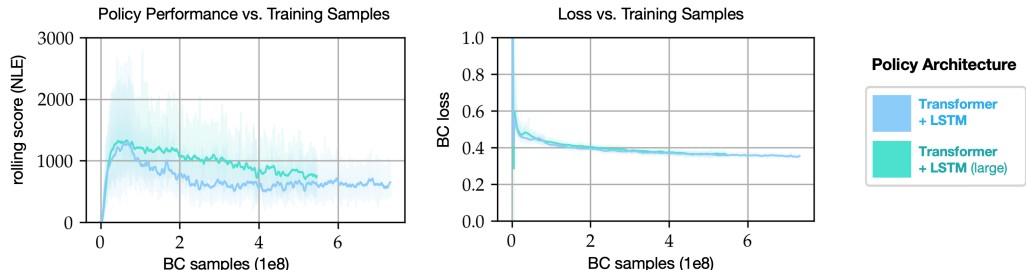

**Model Parameter Scaling: From 50.8M to 98.6M Parameter (Transformer + LSTM) Policies**

Figure 11: **Model Parameter Scaling experiment training curves**. In each plot, solid lines reflect point-wise averages across 6 random seeds, while shaded regions reflect point-wise min-to-max value ranges across seeds. *Left*: Rolling NLE evaluation score vs total BC training samples, for "default" and 2x deeper transformer-LSTM policies. *Right*: BC loss vs total BC training samples.

# G   Training Curves

We provide training curves reflecting all conducted experiments. In Figures 11 and 12, we display both *rolling NLE scores* as well as BC loss curves as a function of training samples, across all model and data scaling experiments presented in Section 5. In Figure 13, we display aggregate rolling NLE scores as a function of training samples for all remaining BC and APPO + BC experiments, separated according to model family.

**Rolling NLE Score**   The rolling NLE score metric introduced and displayed in the figures discussed here reflects an evaluation of policy performance on withheld NLE instances conducted continually during model training in a "rolling" fashion via a fixed number of CPU workers. As such, this metric is biased towards shorter-length games, with the value of smoothed rolling score as a function of

---

[5]As indicated by overall mean NLE score in the "standard evaluation" procedure.

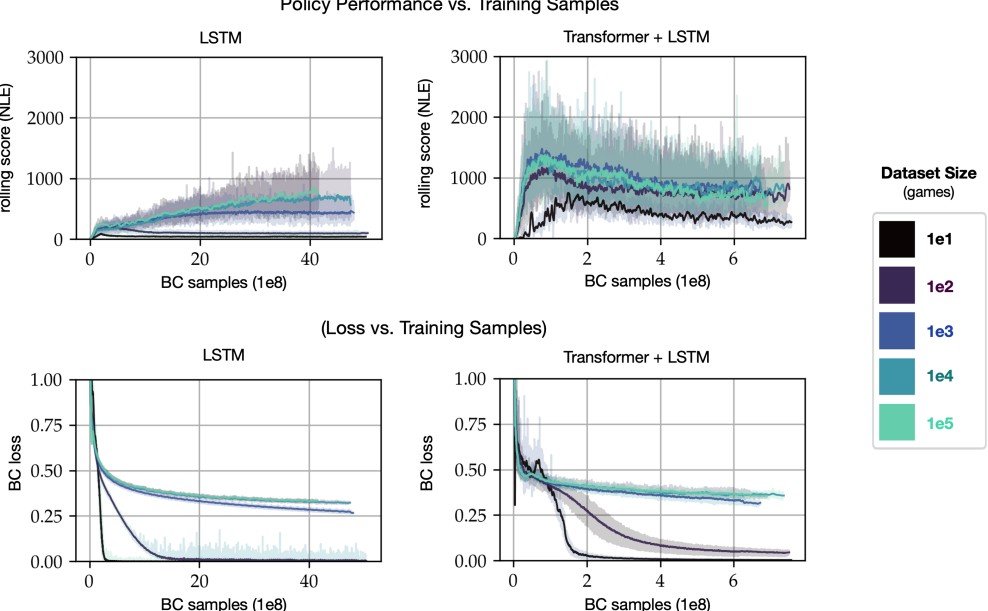

Figure 12: **Dataset scaling experiment training curves**. All experiments for a given dataset size were trained on a dataset sub-sampled without replacement from `HiHack`. The sub-sampling procedure was seeded. Training hyperparameters and model architectures were identical across all runs belonging to a single policy class. As in Figure 12, solid lines reflect point-wise averages across 6 random seeds, while shaded regions reflect point-wise min-to-max value ranges across seeds in all plots. *Top*: Rolling NLE evaluation score vs total BC training samples, across dataset sizes for the "LSTM" and "transformer-LSTM" policy classes. *Bottom*: BC loss vs total BC training samples, across dataset sizes for the "LSTM" and "transformer-LSTM" policy classes.

training sample confounded by the policy-specific relationship between NLE score and total game turns. Rolling NLE score is thus *not* interchangeable with the large-batch "standard evaluations" employed elsewhere in this paper and presented in-depth in Appendix F. However, unlike BC loss, it serves as an efficient and useful (if noisy) proxy measure of policy generalization over the course of training.

**Model Parameter Scaling Training Curves**  As shown in Figure 11, the generalization capability of transformer-LSTM policies (approximated via rolling NLE score) peaks soon after the start of training, decaying and flattening out as training proceeds despite continued improvement in BC loss. This observation supports the claim made in Section 7 that transformer based models overfit to `HiHack`.

Despite the aforementioned noisy nature of rolling NLE score, the "Policy Performance vs. Training Samples" curves on the left of Figure 11 allude to our large-scale "standard evaluation" finding from Section 5; namely, that policy performance does not increase when model parameter count is scaled up, even after training hyperparameters are tuned. Similarly, the "Loss vs. Training Samples" curves on the right of this figure indicate nearly identical training errors across both models as a function of BC training samples.

**Dataset Scaling Training Curves**  In Figure 12, we note that for datasets consisting of, or exceeding, 1000 `AutoAscend` games, LSTM policies do not appear to overfit over the course of our BC experiments, with rolling NLE score consistently monotonically increasing as a function of BC training samples for all such policies. However, a positive relationship between dataset size and maximal rolling NLE score over training persists, indicating that the addition of more data does lead to measurable (if sub log-linear) improvements in policy generalization. An inspection of

LSTM policy loss curves suggests a similar story. Losses across policy seeds trained on 10 and 100 `AutoAscend` games drop swiftly to values near zero, supporting this overfitting hypothesis.

The transformer-LSTM policy loss curves in Figure 12 also reveal harsh overfitting for policies trained with 10 and 100 games. Interestingly, the generalization capability of these policies, again indicated by rolling NLE score over training, is vastly superior to that of their LSTM counterparts. Indeed, we observe that a transformer-LSTM policy trained on just 100 games vastly outperforms a pure LSTM policy trained on 10x as many games. We attribute this gap to the frozen nature of the pre-trained LSTM component of the transformer-LSTM policies employed as a recurrent encoder in these policies, and hypothesize that it is the static quality of the recurrent representation output by this encoder which bolsters the generalization capability of the resultant models in exceptionally low-data regimes.

Figure 13: **Aggregate rolling NLE evaluation score curves for the central BC and APPO + BC experiments discussed in this paper**. Model parameter and dataset scaling training curves are omitted here on account of being visualized in Figures 11 and 12, respectively. For APPO + BC experiments, batch accumulation was employed to ensure that the ratio of RL to BC samples "seen" during training was 1:1. This property is indicated via the secondary x-axes of APPO + BC figures here, which show RL sample quantities. *Top*: Rolling NLE evaluation score vs total BC training samples in pure BC experiments, across non-hierarchical and hierarchical LSTM and transformer-LSTM based policy classes. *Bottom*: Rolling NLE evaluation score vs total BC training samples in APPO + BC experiments, across non-hierarchical and hierarchical LSTM and transformer-LSTM based policy classes.

**Aggregate BC and APPO + BC Training Curves**    The "Aggregate Policy Performance vs Training Samples" curves of Figure 13 align with the general model family training trends previously observed in the scaling experiments. Notably, we find once again that LSTM based policies' rolling NLE scores improve monotonically with training samples whether training is conducted with BC or APPO + BC, while this is not the case for transformer-LSTM based models. Indeed, the generalization properties' of policies belonging to this model family improve at the start of training before worsening as training proceeds across BC experiments. We interpret continued demonstration of these trends as further support for the claims of LSTM underfitting and transformer-LSTM overfitting made in Section 7 of the main body of the paper as well as in Appendix F.

Moreover, we note that the introduction of an RL loss induces a particularly large amount of volatility in the rolling NLE score associated with transformer-LSTM based models, disrupting the monotonically decreasing relationship between score and samples previously observed for these policies following $\approx 2 \cdot 10^8$ training samples in the pure BC experiments (i.e. the eventual overfitting behavior). This observation suggests that the performance of policies belonging to this model family is bottlenecked by an insufficient total throughput of "on-policy" or interactive data.

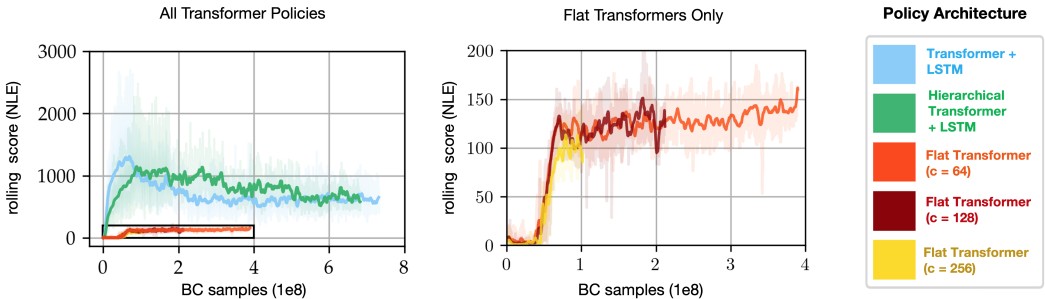

Figure 14: **Aggregate rolling NLE evaluation score curves for flat transformer policies vs transformer-LSTM and hierarchical transformer-LSTM policies during BC training**. Flat transformer policies have similar architecture to transformer-LSTMs with two exceptions: (1) the pretrained, frozen LSTM component of the latter is ablated from the former; and (2) the core modules of tested flat transformer policies consist of twice as many (i.e. 6) transformer encoder layers . We perform a sweep over values of the context-length parameter $c$, corresponding to the number of consecutive timesteps included in the flat transformer context length per sample during training with BC. A total of six random seeds for each model class were employed to test flat transformer policies. Training for each random seed was conducted on a single GPU for 48 hours.

**Ablating the LSTM Component of transformer-LSTM Policies**   To establish and confirm the importance of the recurrent LSTM component of the transformer-LSTM policies introduced in this paper, we perform a set of ablation experiments with policies featuring a transformer core module only, meaning all input is confined strictly to the transformer's context window at each iteration of training and inference. We refer to models with such an architecture as *flat transformers*. As demonstrated in Figure 14, we find flat transformers to perform substantially worse in evaluation when trained with behavioral cloning on AutoAscend demonstration data than either their pure LSTM or transformer-LSTM counterparts. The context-length parameter, denoted as $c$ in the legend of Figure 14, reflects the number of contiguous observations employed for action prediction. We perform a small sweep over values of this parameter in an effort to establish the relationship between observation history and model generalization. Reported results reflect the best performing flat transformer architecture after conducting hyperparameter tuning across transformer encoder layer size, layer count, and attention head count.

We make two key observations: (1) the number of samples seen during the 48-hour training window is inversely proportional to the context length employed in training; (2) the rate of BC policy improvement does not increase with context length, across the set of context length values tested here. It is on this basis of (2), coupled with the fact that the average AutoAscend demonstration in our HiHack dataset consists of 27000 keypresses (Table 1), several orders of magnitude beyond the longest context length, $c = 256$, we find feasible to test for flat transformers, that we hypothesize that an encoding of full game history is important for imitation in our setting.

# H   Low-Sample Hypothesis Testing

To establish the statistical significance of the core BC and APPO + BC empirical results reported in the main body of this paper, we perform a series of two sample t-tests with a decision threshold of $\alpha = 0.05$. The results of these t-tests are summarized in Tables 7, 8, and 9. We confirm all reported findings.

Table 7: **Statistical significance of policy comparisons across BC experiments**: two-sample t-tests on mean policy NLE scores $\overline{s}$ in evaluation, across six random seeds (df = 5). We compute p-values associated with three different null hypotheses, testing for improvement in performance/generalization over the baseline LSTM (i.e. `CDGPT5` [23]) and hierarchical LSTM policies, as well as a decline in performance/generalization over the 3-layer non-hierarchical transformer-LSTM for models trained with BC.

| | Hierarchy | $H_0: \overline{s}_? \leq \overline{s}_{\text{LSTM}}$ | | $H_0: \overline{s}_? \leq \overline{s}_{\text{hier-LSTM}}$ | | $H_0: \overline{s}_? \geq \overline{s}_{\text{trnsfrmr-LSTM}}$ | |
|---|---|---|---|---|---|---|---|
| | | p-value | reject $H_0$ | p-value | reject $H_0$ | p-value | reject $H_0$ |
| LSTM (XXL dec) | ✗ | .320265 | ✗ | - | - | - | - |
| LSTM | ✓ | .000047 | ✓ | - | - | - | - |
| Transformer-LSTM | ✗ | < .00001 | ✓ | .000125 | ✓ | - | - |
| Transformer-LSTM (large) | ✗ | .000014 | ✓ | .001646 | ✓ | .001792 | ✓ |
| Transformer-LSTM | ✓ | .000063 | ✓ | .002549 | ✓ | .016935 | ✓ |

Table 8: **Statistical significance of policy comparisons across APPO + BC experiments**: two-sample t-tests on mean policy NLE scores $\overline{s}$ in evaluation, across six random seeds (df = 5). As in Table 7, we compute p-values associated with three different null hypotheses, but across model classes trained with APPO + BC.

| | Hierarchy | $H_0: \overline{s}_? \leq \overline{s}_{\text{LSTM}}$ | | $H_0: \overline{s}_? \leq \overline{s}_{\text{hier-LSTM}}$ | | $H_0: \overline{s}_? \geq \overline{s}_{\text{trnsfrmr-LSTM}}$ | |
|---|---|---|---|---|---|---|---|
| | | p-value | Reject $H_0$ | p-value | Reject $H_0$ | p-value | Reject $H_0$ |
| LSTM (XXL dec) | ✗ | .470427 | ✗ | - | - | - | - |
| LSTM | ✓ | .001397 | ✓ | - | - | - | - |
| Transformer-LSTM | ✗ | .042454 | ✓ | .348223 | ✗ | - | - |
| Transformer-LSTM | ✓ | .026776 | ✓ | .362569 | ✗ | .102058 | ✗ |

Table 9: **Statistical significance of gains in policy performance from APPO + BC vs pure BC**: two-sample t-tests on mean policy NLE score $\overline{s}$ in evaluation, across six random seeds (df = 5). We compute p-values associated with the null hypothesis that policy performance yielded by APPO + BC training is less than or equal to that yielded by pure BC. We reject this null hypothesis for all but non-hierarchical transformer-LSTM models, for which we see no statistically significant improvement in policy generalization with RL fine-tuning.

| | Hierarchy | $H_0: \overline{s}_{?(\text{APPO + BC})} \leq \overline{s}_{?(\text{BC})}$ | |
|---|---|---|---|
| | | p-value | Reject $H_0$ |
| LSTM [23] | ✗ | .000121 | ✓ |
| LSTM (XXL dec) | ✗ | .000017 | ✓ |
| LSTM | ✓ | .000066 | ✓ |
| Transformer-LSTM | ✗ | .416954 | ✗ |
| Transformer-LSTM | ✓ | .003269 | ✓ |

# I Distributional Visualizations of Evaluation Results

**Max Dungeon Level Reached vs. Total Turns**  In Figure 15, we supplement the absolute mean and median max-dungeon level and agent life-time (in game turns) statistics introduced in Table 2 with 2-D distributional visualizations of both the raw values of these metrics as well as their mutual inter-relationship, evaluated via $n = 3402$ seeded NLE games run individually for all policy class representatives in our "in-depth evaluation."

The `AutoAscend` 2-D contour plot on the right of this figure reveals an interesting emergent property of the bot's behavior, which is confirmed by study of individual games: the high-level "descent behavior" of `AutoAscend` appears to fall along one of two modes. In the first of these modes, the bot spends a very large amount of turns on dungeon level 1, and avoids descending further into the dungeon before the end of the game. In the second mode, the bot begins to rapidly descend fairly deep into the dungeon, reaching as far as level 11 in the sample of game seeds tested here. The overall

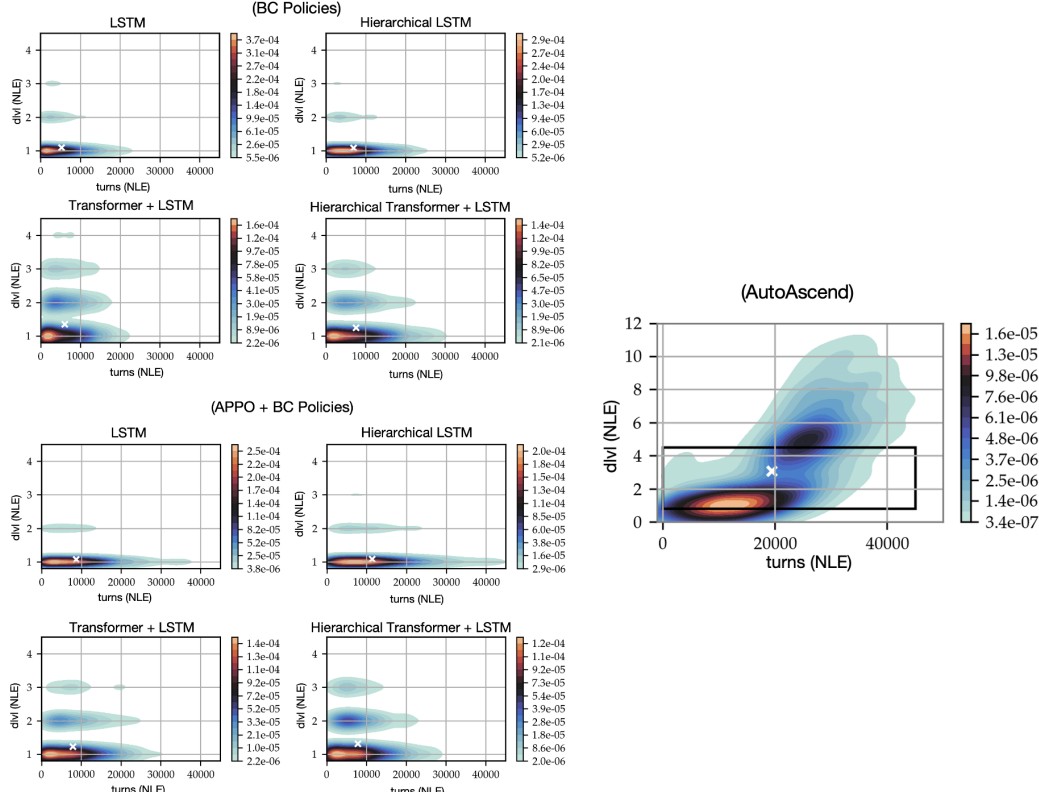

Figure 15: **Max dungeon level reached vs. total turns across "in-depth evaluation" games, visualized via 2-D contour density plots**. Contour densities are indicated by the color-bars accompanying each subplot. Mean quantity values (computed dimension-wise for all policies) across the "in-depth evaluation" batch are indicated by white '✗' symbols in each subplot. The symbolic bot's vastly superior ability to play longer and descend much further into the dungeon creates a separation in scale between it and its neural counterparts; as a result, for clarity, we indicate the max dungeon level vs turns subspace displayed in the neural policy contours with a bolded black rectangle in our visualization of `AutoAscend`'s behavior. *Top Left*: Best-performing BC neural policy seeds. *Bottom Left*: Best-performing APPO + BC neural policy seeds. *Right*: `AutoAscend`.

relationship between `AutoAscend`'s total in-game life-time and max-dungeon level reached appears to be roughly quadratic.

In contrast, none of the evaluated neural policy class representatives comes close to achieving in-game progress resembling `AutoAscend`'s secondary behavior mode. A large majority of games for all neural policy classes appear to end on the first level of the dungeon, with policies very rarely surviving as long on this level as `AutoAscend`. This suggests that neural policies may be failing to master the long-range, very low-level behaviors of the bot, even when these behaviors are factorized across `AutoAscend` strategies, as in the case of hierarchical policy variants.

Nevertheless, the qualitative performance of hierarchical policies clearly improves upon that of non-hierarchical models, with the hierarchical transformer-LSTM policy trained with BC both surviving longer of dungeon level 1 and descending with higher frequency than other BC-trained policy representatives. Furthermore, this qualitative behavior appears to be strengthened when interaction is added into the mix, with the mode centered on "dungeon level 2" increasing in density for the APPO + BC variant of the hierarchical transformer-LSTM representative policy.

Taken together, these sets of observations lead us to hypothesize that the quality of neural policies trained with imitation learning on extremely complex, long-horizon tasks like NetHack may be

further improved with increases in the scale of hierarchically-informed, fixed behavioral factorization, beyond that explored in this paper. We consider this to be a very exciting direction for future work.

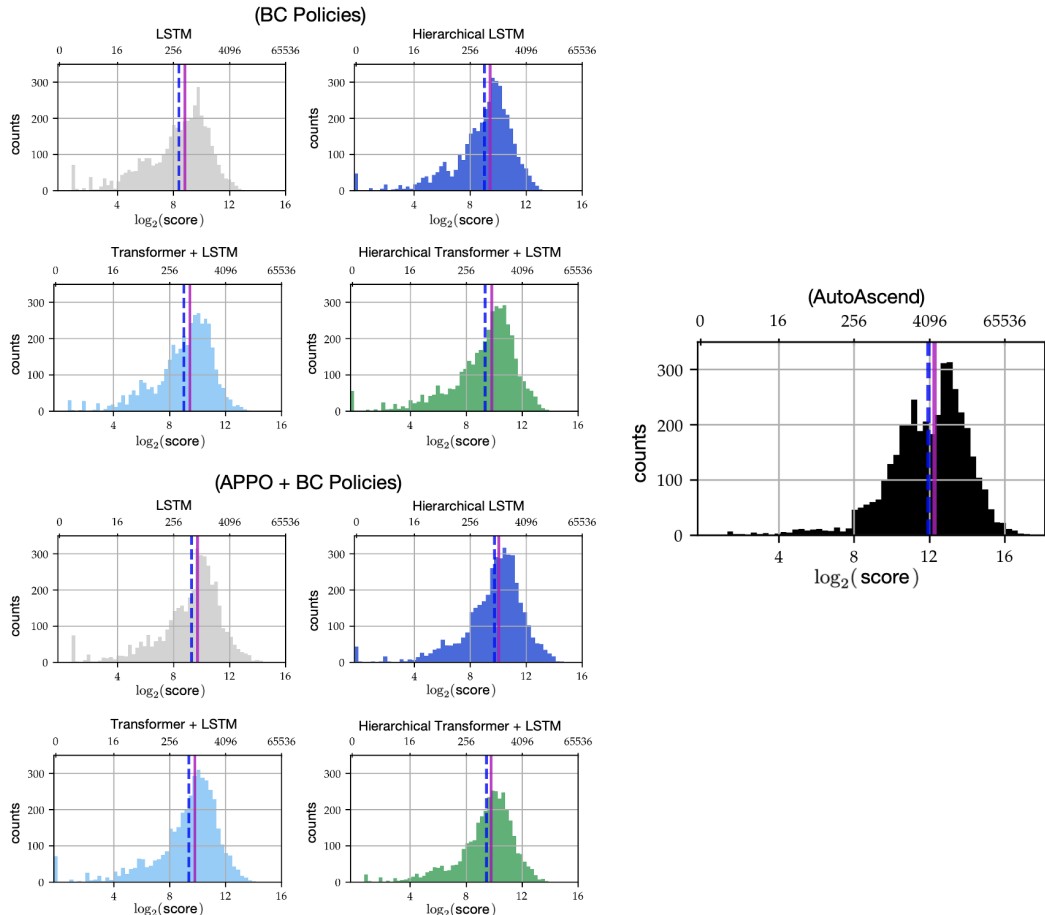

Figure 16: **Log-score distribution across "in-depth evaluation" games**. As in Figures 8 and 10, we indicate absolute median and mean values of policies' NLE scores in the "in-depth evaluation" with dashed-blue and solid-pink lines. The introduction of an RL loss reduces the "mass" associated with the left-tail of log-score and increases the "mass" associated with the right-tail across all neural policy classes, inducing right-ward shifts in median and mean scores, though difficult to perceive in this figure on account of the log-scale of the x-axis. We refer the reader to Table 2 in the main paper for aggregate absolute numerical values of NLE score. *Top Left*: Best-performing BC neural policy seeds. *Bottom Left*: Best-performing APPO + BC neural policy seeds. *Right*: `AutoAscend`.

**Log-Score Distributions**   We conclude this supplementary analysis with a visualization of log-score distributions across neural policy classes in Figure 16, computed over the "in-depth evaluation" seeded games. The trends displayed in this figure align with those described and demonstrated previously. Improvements in model architecture as well as the introduction of hierarchy and interactive learning lead to a re-distribution of "mass" between left and right tails of distribution, with the overall counts of "low score" games decreasing and "high score" games increasing when these improvements are applied. The gap to `AutoAscend` is significantly reduced, but not bridged.

