# OpenReview forum: "NetHack is Hard to Hack"
_NeurIPS.cc/2023/Conference — NeurIPS 2023 poster_

### Official Review · Reviewer_83YB · 2023-07-04

**Soundness:** 3 good
**Presentation:** 4 excellent
**Contribution:** 2 fair
**Rating:** 6
**Confidence:** 4

**Summary:**

This paper seeks to study and better understand the large performance gap between neural and symbolic agents in the NeurIPS 2021 NetHack Challenge. The main hypothesis is that symbolic agents advantage derives from hierarchical reasoning, which was not an element in participating neural agents. To test this hypothesis, a new dataset is generated using the winning symbolic agent that records both actions and higher-level strategic labels and a neural behavior cloning agent was trained with this augmented data. Beyond this, the paper also explores the impact of increased model size and dataset size, changes to the neural architecture, and the addition of a policy fine-tuning step using a reinforcement learning algorithm. The results suggest that hierarchical training improves the neural agent's performance significantly more than increased model capacity but that more powerful model architectures (i.e. a transformer-based model) could overfit to the augmented data.

**Strengths:**

Quality/Soundness

The hypotheses and claims of the paper were laid out clearly and the experiments were well-designed to evaluate them.

Clarity/Presentation

I found the paper to be clearly presented and well-written. At each stage I had a clear understanding of the question under investigation and the methodology for studying it.

Originality/Contribution

The paper is explicit that its main contribution is not algorithmic but scientific. Since the scientific questions posed here are grounded in the performance of agents from a specific competition that happened last year, I expect that this analysis is original.

Significance/Contribution

Increasing understanding of the performance gap in the NetHack problem, as a proxy for complex, long-horizon problems in general, could be important. Symbolic approaches make use of quite a lot of domain expertise applied to constructing the symbolic structure, making them effective in their target problem but inflexible. Neural networks seem to be quite flexible and capable of learning without a lot of structure engineering, but don't seem to be able to take advantage of structure in the environment. It seems sensible to study the primary factors preventing neural approaches from building the same long-range structures.

I think it is worthwhile to see the comparison between this structural change and the alternative interventions of more data, more parameters, or more sophisticated/expensive architecture. The finding that, in a time-constrained setting (true of many decision-making problems), model structure may be more important than model capacity seems likely to at least spark interesting conversations within the community. The fact that there is plenty of performance gap still to cover, even with this built-in domain knowledge may also inspire further investigation into what measures could come closer to closing the gap and how those insights might be applied to more general practice.

Overall

Overall, I find this to be a clearly written paper with a reasonable scientific question and sound methodology to address it (modulo some missing statistical analyses). The findings are not revolutionary but, to my eyes, they do provoke further questions about neural approaches to learning in complex, long-horizon problems and may inspire follow-up work either in the NetHack testbed specifically or in studying these questions more generally.

**Weaknesses:**

Quality/Soundness

My main concern in this area is the small number of independent samples per model class (6). I do understand that these results are generated at great expense and that it may not be feasible to generate more trials, but the small number of samples diminishes the statistical power of these analyses. I can see that the error bars are quite small; assuming those are showing the standard error, that's encouraging. Nevertheless, whether more samples can be generated or not, I think it's important that the paper include the findings of low-sample hypothesis testing (e.g. t-test) on these results; without that, we can't confidently distinguish between noise and meaningful differences.

Originality/Contribution

The paper acknowledges that behavior cloning, hierarchical policies, and transformer-represented policies have all been studied in prior work.

Significance/Contribution

NetHack is not, in and of itself, an intrinsically important problem to solve.

The results presented here are not conclusive or enormously surprising. The main result is fairly predictable: adding explicit supervision about high-level strategy and explicit hierarchical structure in the model helps the model take advantage of hierarchical structure in the environment.

Overall

Overall, I find this to be a clearly written paper with a reasonable scientific question and sound methodology to address it (modulo some missing statistical analyses). The findings are not revolutionary but, to my eyes, they do provoke further questions about neural approaches to learning in complex, long-horizon problems and may inspire follow-up work either in the NetHack testbed specifically or in studying these questions more generally.

---After discussion---

I have considered the other reviews and the authors' responses. I continue to feel confident about my overall assessment.

**Questions:**

n/a

**Limitations:**

Since the paper does not propose significantly new algorithmic ideas, the main source of limitations would be in the methodology and analysis. I generally found the paper to avoid overclaiming. I've already discussed one area where this aspect of the paper could be improved: acknowledgement of the small sample sizes and proper statistical analysis to inform the conclusions. The other area might be in the conclusions where perhaps the summary of the findings might be a bit too general and could be toned down and/or stated clearly as hypotheses (e.g. "Hierarchy hurts overfitting models" is an overly broad conclusion from a limited set of experiments but seems like a reasonable hypothesis given these results).

---

> ### Author Rebuttal · Authors · 2023-08-10
>
> Dear Reviewer,
>
>
> Thank you for the time and effort that you have taken to review our submission. We are delighted that you found our paper to be well-written and the insights generated by our experiments to be valuable as a basis for future work on neural approaches to learning in complex, long-horizon environments.
>
>
> We hope to address your remaining concerns and questions below.
>
>
> > My main concern in this area is the small number of independent samples per model class (6).
>
>
> Thank you for pointing out our omission of a clarification on the meaning of the error bars plotted in the results figures of our draft submission; indeed, the error bars in Figures 3, 4, and 5 all indicate one standard error across policy random seeds. The absolute numerical values of evaluation metrics' standard errors are also reported in Table 2, which aggregates all of the experimental results of the paper. We will remedy this omission in the revised version of our submission.
>
>
> As you note, due to the great computational expense associated with trials, we regret that we are unable to increase the number of independent trials run per model and experiment class in this paper.
>
>
> We agree that a thorough verification of the statistical significance of our results is important. We will include the low-sample hypothesis tests that you suggest in the revised version of this paper.
>
>
> In the interim, please note that the values of the key metric we employ to compare model performance on held-out seeds of NLE (mean score across a fixed set of 1024 unique, procedurally-generated NetHack games) are separated by two standard errors for all model classes in both our BC and APPO + BC sets of experiments.
>
>
> > NetHack is not, in and of itself, an intrinsically important problem to solve.
>
>
> We concur that NetHack may not be an intrinsically important problem to solve. Nevertheless, we respectfully contend that this does not diminish the importance of work employing this environment as a general testbed for studying flexible policy learning in long horizon, open-ended environments.
>
>
> Please also take a look at **section (i)** of our general rebuttal.
>
>
> > The main result is fairly predictable: adding explicit supervision about high-level strategy and explicit hierarchical structure in the model helps the model take advantage of hierarchical structure in the environment.
>
>
> To our knowledge, our paper is the first to demonstrate that hierarchical labels can offer a stable and powerful mechanism for embedding behavioral priors into policies during pre-training.
>
>
> As observed by reviewer qyRC, currently, there are few complex environments where expert demonstrations are accompanied by hierarchical labels, though such labels are in-principle no more difficult to collect than the demonstrations themselves from annotators.
>
>
> Though we agree that our findings concerning the power of hierarchy may not be surprising from a theoretical perspective, we are hopeful that our paper may spur the creation of more datasets with this property as well as further investigations of how hierarchical labels may be best exploited for offline and online learning of flexible neural policies, both in NetHack and beyond.
>
>
> > The other area might be in the conclusions where perhaps the summary of the findings might be a bit too general and could be toned down and/or stated clearly as hypotheses.
>
>
> Thank you for these suggestions, we will tone down the language employed in this section of the paper accordingly.
>
>
> Once again, we are sincerely grateful for your feedback on our submission. Please let us know if you have any outstanding comments, questions, or concerns preventing a stronger recommendation for acceptance.

---

> > ### Comment · Reviewer_83YB · 2023-08-12
> > **Response to rebuttal**
> >
> > Thanks for your response! I think we are generally in agreement about both the value of using NetHack and the limitations of gathering findings in a single domain. Similarly, I think we generally agree about the potential illustrative value of the results and the dataset as well as the limited surprise factor in the findings.

---

### Official Review · Reviewer_qyRc · 2023-07-06

**Soundness:** 3 good
**Presentation:** 3 good
**Contribution:** 3 good
**Rating:** 7
**Confidence:** 4

**Summary:**

# Problem Statement
The paper addresses the challenge of neural policy learning methods struggling in long-horizon tasks, particularly in open-ended environments with multi-modal observations, such as the game NetHack. It was observed that symbolic agents significantly outperformed neural approaches in the NeurIPS 2021 NetHack Challenge.

# Main Contribution
The paper's main contribution is an extensive study on neural policy learning for NetHack. The authors analyzed the winning symbolic agent and extended its codebase to generate one of the largest available demonstration datasets. They examined the advantages of an action hierarchy, enhancements in neural architecture, and the integration of reinforcement learning with imitation learning. Their investigations resulted in a state-of-the-art neural agent that surpassed previous fully neural policies by 127% in offline settings and 25% in online settings on median game score. However, they also demonstrated that mere scaling is insufficient to bridge the performance gap with the best symbolic models or even the top human players.

# Methodology and Experiments

## The Hierarchical HiHack Dataset
The authors create the HiHack dataset, which is a hierarchically-informed version of the NetHack Learning Dataset (NLD-AA), containing 3 billion recorded game transitions from over a hundred thousand games played by the AutoAscend agent.

## Hierarchical Behavioral Cloning
The authors extend the ChaoticDwarvenGPT5 (CDGPT5) model, a top-performing open-source neural model for NetHack, by introducing a hierarchical decoding module. The model consists of three separate encoders for different types of observations and an LSTM core module. The hierarchical extension replaces the linear decoder of the CDGPT5 model with a hierarchical decoder that predicts the strategy label and selects the appropriate low-level MLP for action prediction. The hierarchical LSTM policy and the baseline non-hierarchical LSTM CDGPT5 policy are trained using a simple cross-entropy loss. The results show that the introduction of hierarchy significantly improves the performance of LSTM policies trained with behavioral cloning, yielding a 40% gain over the baseline in mean NLE score and a 50% improvement in median score across seeds. The authors confirm that this improvement is due to hierarchy and not simply a result of the increased parameter count of the hierarchical LSTM policy.

## Architecture and Data Scaling
The authors explored scaling as a potential solution to improve the performance of the model, which was significantly behind the symbolic policy used to generate the HiHack demonstrations. They developed a novel base policy architecture for NetHack that introduces a Transformer module into the previous CDGPT5-based architecture. They also conducted data scaling experiments using subsets of the HiHack dataset to examine the relationship between dataset size and the test-time performance of BC policies. The results showed that both the non-hierarchical and hierarchical variants of the combined transformer-LSTM policy architecture yielded gains, but the larger model performed worse than the smaller one due to overfitting. This suggested that scaling of model capacity alone would not be sufficient to close the neural-symbolic gap. Additionally, brute force scaling of the dataset alone could not viably close the gap to symbolic methods.

## Combining Imitation with Reinforcement Learning
The authors explored combining imitation learning with reinforcement learning (RL) to bridge the performance gap with AutoAscend. They used a combination of behavioral cloning (BC) and asynchronous proximal policy optimization (APPO) for training. The results showed that RL fine-tuning significantly improved the performance of all models. The best-performing approach was APPO + BC using the hierarchical LSTM model, which achieved a new state-of-the-art for neural policies on NLE, surpassing the previous best result by 48% in mean NLE score and 25% in median NLE score. The Transformer-LSTM models performed worse due to their slower training speed and the fixed training time budget. The authors also observed that fine-tuning with RL improved the error-correction capability of models across all model classes compared to their purely offline counterparts.

**Strengths:**

# Originality
The problem is interesting and the approaches are insightful.

# Quality
The analysis and experiments are comprehensive.

# Clarity
The article is overall well written and clear.

**Weaknesses:**

1. The current focus of the study is quite narrow, being primarily centered on the application of imitation learning for NetHack, limiting its influence. In the context of mastering the game, while this approach is interesting, it is unlikely to exceed the performance of experts that generate demonstrations, not to mention that the experts are already algorithms that can scale well. Furthermore, NetHack, despite being an excellent game, is somewhat niche and its real-world implications are relatively minimal. The techniques proposed in this study are specifically tailored for this game, which limits their potential for inspiring more universally applicable methods that could have a broader impact.
  - The availability of hierarchical labels is a strong assumption that does not often hold, which further limits the applicability of the proposed methods.

2. Even just for bridging the performance gap between neural models and AutoAscend, there is no promising direction revealed by the work as the various augmenting components seem to contradict each other.

**Questions:**

1. When introducing Transformer to augment the capacity of the neural model, why did authors choose the architecture as shown in the article? Specifically, transformers are best known for their NLP and CV capacity, which could make them good replacement for the CNN and MLP encoders.
2. Why do the authors enforce the 48 hour training time cap instead of training all models till convergence? Given that this study does not appear to prioritize data efficiency or training efficiency, the necessity of such a computational time constraint is unclear. It would be beneficial to understand the rationale behind this choice, as it may not directly align with the study's primary objectives.

**Limitations:**

The authors note that possible avenues for future exploration include: (a) methods for increasing the Transformer context length to give the agent a longer memory to aid exploration; (b) addressing the multi-modal nature of the demonstration data (i.e. quite different trajectories can lead to the same reward), which is a potential confounder for BC methods. Some forms of distributional BC (e.g. GAIL, BeT) could help alleviate this issue.

The aforementioned two points do not address the limitations raised in the "Weakness" section.

---

> ### Author Rebuttal · Authors · 2023-08-10
>
> Dear Reviewer,
>
>
> We thank you for your thoughtful and detailed feedback on our submission. We look to address your remaining concerns about our paper below.
>
>
> > ...while this approach is interesting, it is unlikely to exceed the performance of experts that generate demonstrations, not to mention that the experts are already algorithms that can scale well.
>
> Rather than to achieve SOTA in NetHack, our goal in this paper is to employ NLE as a testbed for investigation of the general performance gaps between neural policies and symbolic agents in long-horizon, open-ended environments.
>
> Please also note that while AutoAscend is the SOTA artificial agent in NLE, the median in-game score it achieves is still several orders of magnitude short of a typical expert player’s “ascension” score. The game remains unsolved by any artificial agent [1].
>
>
> >  The techniques proposed in this study are specifically tailored for this game, which limits their potential
>
> Please take a look at **section (i)** of our general rebuttal.
>
>
> > The availability of hierarchical labels is a strong assumption that does not often hold, which further limits the applicability of the proposed methods.
>
> Hierarchical labels have been gathered for other settings, such as in the 3D DeepMind Playhouse environment, where human annotators were asked to provide natural language instructions or questions labeling manipulation "subtasks" [2]; thus, they are not impractical to gather. In general, we hope that the impact of hierarchical priors via structured labels revealed by our work might spur the future collection of more demonstration data with this property in open-ended environments.
>
>
> > ...there is no promising direction revealed by the work as the various augmenting components seem to contradict each other.
>
> Our experimental results suggest that the introduction of structured labels is strongly beneficial to the improved performance of lighter-weight, data-limited LSTM policies, both in the BC and BC + RL settings.
>
> However, in the pure BC setting, we find that architectural improvements and scaling make these beneficial effects of structured labels obsolete.
>
> Furthermore, we find a “no free lunch” effect to hold for large transformer-based policies in our BC + RL experiments: they are far more unwieldy to finetune with RL than their smaller LSTM counterparts, with the increased cost of gradient updates dealing a large blow to sample-hungry RL under the compute-time cap. As a result, while we do find interaction to improve the generalization performance of both hierarchical and non-hierarchical transformer-LSTM policies, the performance of these policies on withheld NLE instances is superseded by that of hierarchical LSTMs in this case.
>
> Evaluation data supporting the above is aggregated in Table 2 of the paper.
>
> There are several directions for future work that we find compelling:
>
> (1) Deeper explorations of the impacts of structured labels, both in offline and online settings. Do performance improvements grow as the granularity of hierarchical priors used in pre-training increases? Can we further exploit hierarchical behavioral priors for structured exploration?
>
> (2) Investigation of even better transformer-based policies. If we are able to eliminate the overfitting effects seen in pre-training, does these models’ performance benefit from structured labels too?
>
>
> > When introducing Transformer to augment the capacity of the neural model, why did authors choose the architecture as shown in the article?
>
> Please refer to **section (ii)** of our general rebuttal. As stated above, we will amend our submission to include a discussion of the LSTM ablation upon revision.
>
> > Why do the authors enforce the 48 hour training time cap instead of training all models till convergence?
>
> Constrained model comparisons are common in the computer vision and reinforcement learning literature [3, 4, 5].
>
> We choose to employ a 48-hour training time cap in our study for two reasons.
>
> (1) A compute-time cap **provides a basis for comparison of different neural policy architectures on equal footing**, revealing, for instance, the degree of the performance advantage that faster gradient updates grant LSTM models against bulkier transformer-LSTMs, both in the presence and absence of structured labels. As noted by reviewer 83YB, our finding that in the “...time-constrained setting (true of many decision-making problems), model structure may be more important than model capacity seems likely to at least spark interesting conversations within the community.” Additionally, we believe that a compute-time cap increases the broader impact of our work by making our insights more relevant to researchers constrained by cost, e.g. in academic research settings.
>
> (2) From a practical perspective, employing a training time cap as our comparison basis also **enables us to expand the number of random seeds we employ to report results** across the numerous experiments within the paper, strengthening the statistical significance of our findings.
>
> Thank you once more for the time you have taken to review our paper. Please let us know if you have any outstanding comments, questions, or concerns about our work.
>
>
>
>
> **Citations**
>
>
> [1] Hambro, Eric, et al. "Insights from the neurips 2021 nethack challenge." NeurIPS 2021 Competitions and Demonstrations Track. PMLR, 2022.
>
> [2] Team, DeepMind Interactive Agents, et al. "Creating multimodal interactive agents with imitation and self-supervised learning." arXiv preprint arXiv:2112.03763 (2021).
>
>
> [3] Zhai, Xiaohua, et al. "Scaling vision transformers." Proceedings of the IEEE/CVF Conference on Computer Vision and Pattern Recognition. 2022.
>
>
> [4] Yarats, Denis, et al. "Mastering visual continuous control: Improved data-augmented reinforcement learning." arXiv preprint arXiv:2107.09645 (2021).
>
>
> [5] Peebles, William, and Saining Xie. "Scalable diffusion models with transformers." arXiv preprint arXiv:2212.09748 (2022).

---

> > ### Comment · Reviewer_qyRc · 2023-08-20
> > **Thanks for the rebuttal**
> >
> > Thank you for the comprehensive response.
> >
> > Although I admit that comparisons with the training time cap do provide unique insights and practically makes the experiments more computationally manageable, I want to point out that the training efficiency can vary by a large margin depending on the implementation details, which are non-intrinsic to the algorithmic methods and can obscure analysis. Training all models till convergence allows the influence of the architecture and data factors to manifest more clearly, which are the article's main focus, and in my opinion surpasses the benefits enlisted by the authors of using the training time cap.
> >
> > Other than this point, my questions are properly addressed thanks to the authors' rebuttal and I will update my rating from 6 to 7.

---

### Official Review · Reviewer_PtAe · 2023-07-06

**Soundness:** 2 fair
**Presentation:** 3 good
**Contribution:** 3 good
**Rating:** 7
**Confidence:** 5

**Summary:**

The paper improves the existing solutions in the NetHack Learning Environment (NLE). This is done by taking earlier solutions from a competition around NLE, collecting more data with the best available (symbolic) agent, and using that data to improve a neural only solution. The paper provides experiments with imitation learning (with or without RL tuning), larger models, hierarchical memory setup (LSTM + Transformers) as hierarchical behavioural cloning setup, using labels of the newly collected dataset. While there are improvements, it is still below the demonstrator results, which is then studied by scaling the model sizes and amount data collected. Paper concludes by providing the state of the art results in the task, but also noting that scaling alone is not enough to reach the expert demonstrator level (symbolic agent).

**Strengths:**

- Provides more detailed dataset than the previous works (with hierarchical action labels)
- Sets an interesting premise/task for trying to reach the demonstrators' (AutoHack agent) performance with neural solutions.
- Different ablations to try to answer questions (data/model scaling, model architecture with hierarchy)
- Proposed hierarchical approach to imitate the demonstrator agent.

**Weaknesses:**

While I enjoyed reading the paper, overall I think the results are interesting or applicable to most of the NeurIPS audience, even in the limited scope. The paper presents many results and provides some explanations for them, but does not verify these explanations with further experiments. I think proper answers to these issues would be insightful to many, and others could then use these insights in their work (e.g., where the trained agent failed to imitate the demonstrator? What was the cause of poorer performance? Why did bigger model perform worse?). Creating such insight in one environment would be sufficient, as by focusing on a single environment, you can create very specific scenarios to tease out these answers.

- Limited scope of the work: experiments done in a single environment. Most of the paper is framed in a way that this is not a huge issue (e.g., ablations), but proposing new method just for playing NLE has limited impact. If a new method is proposed to generally improve RL/IL performance, it should be tested at least in two distinct environments.
- Limited improvement in the context of SOTA solutions: 2x over the baseline used in the paper with RL and proposed architecture included, but other neural agents in the NetHack Challenge had higher score. To be interesting in terms of performance, it should at least outperform the NetHack Challenge Neural solutions.
- Proposed method is limited in novelty, as evident by the previous work listed in the paper. If the hierarchical BC figured out the hierarchy automatically (or, if it was an emergent behaviour of the model), that would be more interesting.
- Paper outlines some assumptions on why things failed (e.g., "model overfitted" or "learned to self-correct"), but these claims were not verified with results. The paper would be much stronger if you can give solid, verified answer that indeed, overfitting was to blame or that RL trained the model to "self-correct".

**Questions:**

Questions:
1) In multiple occasions paper says that the lower performance of bigger model is due to overfitting (e.g., line 229). However there are no results/experiments to show that this indeed was the case. A simple way to find this out is to do train-validation (or even train/validation/test) split, and testing on held out data as training progresses.
2) Regarding data scaling experiments: did you change any other settings of the training setup when increasing data amount? Previous work has demonstrated that the optimal model size and/or training compute depends on the amount of data (Hoffmann et al. 2020).
3) Regarding model scaling experiments: I assume only the number of layers in the transformer was changed? The bottleneck of the network may be elsewhere, e.g., one of the input layers or output layers. I would recommend scaling the whole network, similar to what OpenAI VPT work did, where ResNet blocks were "widened" in terms of filters, as well as increasing transformer size (Baker et al. 2022). Also, Hoffmann et al. (2020) changed number of layers, number of attention heads and transformer dimensionality when scaling models. This might be something you want to try.
4) Instead of LSTM + Transformer model, did you experiment with transformer model only? E.g., akin to VPT work (Baker et al. 2022), embed all inputs into one vector, stack vectors over timesteps, apply causal transformer, and predict actions from the transformer outputs. This type of model might scale better, as it reduces the amount of components that might interfere.

#### Comments (not questions)
- Fig1 right: weird scale. Any chance to get more points?
- Line 205: grammar error at the start of the line
- Explain/rename "Dlvl" and why "Turns" is good metric
- Figure 3: "LSTM + XXL Dec" is bit confusing naming, since "decoder" is not commonly used term in the paper. I'd recommend using something like "LSTM (bigger)" to simply reflect that it is the LSTM baseline but with bigger network
- Figure 3 (and others): add explanation to caption what is the error bar of the bar plots. Is it standard deviation or standard error (or something else)?
- Table 2 caption: starts with weird "[V4]"

#### References

- Hoffmann, Jordan, Sebastian Borgeaud, Arthur Mensch, Elena Buchatskaya, Trevor Cai, Eliza Rutherford, Diego de Las Casas et al. "Training compute-optimal large language models." arXiv preprint arXiv:2203.15556 (2022).
- Baker, Bowen, Ilge Akkaya, Peter Zhokov, Joost Huizinga, Jie Tang, Adrien Ecoffet, Brandon Houghton, Raul Sampedro, and Jeff Clune. "Video pretraining (vpt): Learning to act by watching unlabeled online videos." Advances in Neural Information Processing Systems 35 (2022): 24639-24654.

**Limitations:**

No explicit sections for limitations or broader/societal impact was given. Authors bring up the future work ideas in the conclusion. While I think the work does not require societal impact section (no immediate impact), I urge authors still think through of any cases where the work or the insights could impact others. Or alternatively, what impact would _not_ including some results do (e.g., skipping some analysis).


## Rebuttal acknowledgement

I have read authors' rebuttal which did address my concerns, and I increased my rating from 4 to 7 to signal my vote to accept this paper (change was done before discussion period closed).

---

> ### Author Rebuttal · Authors · 2023-08-10
>
> Dear Reviewer,
>
>
> We thank you for your detailed and thoughtful feedback on our paper, which has greatly helped to further strengthen our work. We hope to address the concerns and questions that you raise in your review below.
>
>
> > If a new method is proposed to generally improve RL/IL performance, it should be tested at least in two distinct environments.
>
>
> Our goal in this work is to conduct a deep, systematic, and scientific investigation of performance gaps between data-driven neural policy learning and symbolic methods employing NetHack as a testbed, rather than to introduce a new model achieving SOTA RL/IL performance in NLE. As discussed in **section (i)** of the general response above, we respectfully disagree that the insights revealed by our investigations in this paper are limited to NLE.
>
>
> > To be interesting in terms of performance, it should at least outperform the NetHack Challenge Neural solutions.
>
>
> The two neural policies outperforming the CDGPT5 baseline from the NetHack Challenge (NHC) that you are referring to rely on hand-engineered augmentations including separated action spaces, role-specific training, and even hard-coded subroutines. CDGPT5 is not only the best open-sourced neural model from the NHC, but it is also the best purely data-driven model [1].
>
>
> As stated above, our goal in this study is to probe the general capability of neural approaches at learning complex, flexible behaviors directly from multimodal data; hence, we refrain from reliance on any NLE-specific policy augmentations. Respectfully, we believe that this choice makes our analysis more interesting, both from a scientific perspective and to the RL/IL community at large, rather than less.
>
>
> > If the hierarchical BC figured out the hierarchy automatically (or, if it was an emergent behaviour of the model), that would be more interesting.
>
> We thank the reviewer for this suggestion.
>
> To our knowledge, our paper is the first to demonstrate that hierarchical labels can offer a stable and powerful mechanism for embedding behavioral priors into policies during pre-training.
>
> As observed by reviewer qyRC, there are few complex environments where expert demonstrations are accompanied by hierarchical labels, though such labels are in-principle no more difficult to collect than demonstrations. We are hopeful that our paper may spur the creation of more datasets with such labels as well as further investigation of how hierarchical behavioral priors may best be exploited for policy generalization, both in NetHack and beyond.
>
> > In multiple occasions paper says that the lower performance of bigger model is due to overfitting (e.g., line 229). However there are no results/experiments to show that this indeed was the case.
>
>
> We provide detailed support for our claims of model overfitting and underfitting in Section G of the supplementary materials accompanying our paper. In Figures 11, 12, and 13 of this section, we include visualizations of policy performance via “rolling NLE score” [2] on held-out validation seeds of the NetHack Learning Environment (NLE) across training iterations, for all experiments conducted. We will include explicit references to these supplemental figures to help guide readers in the final version of this paper.
>
>
> > ...did you change any other settings of the training setup when increasing data amount?
>
>
> On optimal training compute: The results of the data scaling experiments visualized in Figure 5(right) of the main paper reflect the aggregate, large-scale evaluation performance of “best” model checkpoints across dataset subsample sweeps, selected on the basis of the “rolling NLE score” metric [2]. Consequently, our results in these experiments do reflect provisions for optimal training compute, subject to the 48-hour compute budget that we employ as a basis for fair comparison of policies across model classes. For more details on evaluation procedures, please refer to Section F of the supplementary materials.
>
>
> On optimal model size: The goal of this particular set of experiments was to test the effects of data scaling and model parameter scaling in a decoupled setting; as a result, the model architectures tested in the data scaling experiments were held fixed, with only the size of the dataset varied. Please refer to Section E of the supplementary materials for detailed description of model architectures, with all employed model hyperparameter values included in Tables 5 and 6.
>
>
> > I assume only the number of layers in the transformer was changed?
>
> Indeed, only the number of layers in the transformer was changed here. The architecture of the default transformer-LSTM model (i.e. with 3 layers) was tuned for input and output layer width when these experiments were launched. We will include a version of these experiments with all layer parameters scaled as you suggest in the final version of this submission.
>
> > ...did you experiment with transformer model only?
>
> Please see **section (ii)** of the general response as well as the attached PDF.
>
> > Is it standard deviation or standard error (or something else)?
>
> Thank you for pointing out this omission. Indeed, all error bars on visualizations of experimental results reflect standard error.
>
> We also appreciate your comments identifying typos, and will make according adjustments.
>
> Thank you, once again, for the time you have taken to provide thorough feedback on our submission. We would be grateful if you would notify us whether you have any additional concerns or questions preventing a recommendation for acceptance.
>
> **Citations**
>
>
> [1] Hambro, Eric, et al. "Insights from the neurips 2021 nethack challenge." NeurIPS 2021 Competitions and Demonstrations Track. PMLR, 2022.
>
>
> [2] Hambro, Eric, et al. "Dungeons and Data: A Large-Scale NetHack Dataset." Advances in Neural Information Processing Systems 35 (2022): 24864-24878.

---

> > ### Comment · Reviewer_Ub8t · 2023-08-14
> >
> > Thank you for the comprehensive response!

---

> > ### Comment · Reviewer_PtAe · 2023-08-14
> >
> > Thank you for the extensive comments and additional experiments! The low performance of transformers does indeed make intuitive sense, as with limited context length the agent might miss important information (e.g., state of the inventory). This makes the LSTM + transformer combination way more appealing and potentially very interesting for other applications.
> >
> > With these answers  + comments from other reviews + rethinking, I am increasing my score from 4 to 7 to emphasize my vote to accept this work (avoiding borderline or weaks to clearly signal my vote on this). The argumentation for using NLE alone is valid, although I'd prefer if the work better used this argument to its favor: if we focus on a single environment, we could nitpick on very specific error (or success) cases, and study how different models fail. If one does general "one model to play N games", such nitpicking becomes harder.
> >
> > If accepted, I would suggest authors to:
> >
> > * Open-source the code (just highlighting how important this is, as their work was also based on open-sourced work method).
> > * Include the flat-transformer ("transformer only") results in the main paper and discuss them. This highlights that LSTMs/RNNs have an edge over transformers at least in some cases. The LSTM + transformer architecture, despite being "simple", could potentially be very interesting in other domains.
> > * Highlight the comparison to symbolic-only agents and other neural solutions, as done in the *section (i)* of the general response, better in the paper.
> > * Add the arguments for why you say model is overfitting into the main paper. I still struggle to see how the curves in appendix highlight overfitting; to me it looks more like the model converges. More generally, BC training loss (prediction loss) and model performance are very poorly correlated.

---

### Official Review · Reviewer_Ub8t · 2023-07-12

**Soundness:** 3 good
**Presentation:** 3 good
**Contribution:** 4 excellent
**Rating:** 7
**Confidence:** 2

**Summary:**

This is an emergency review, and I regret that the paper is out of my expertise, which is why my review will rather stay at the surface level.

The paper is concerned with the NetHack challenge, a complex AI challenge that in 2021 reached headlines, because symbolic agents considerably outperformed neural agents. I see three main contributions in the paper:
 - The construction of a large-scale dataset, based on the best symbolic agent and its policy choices, that can enable training better neural agents
 - The training of better neural agents based on this dataset, and other improvements
 - A systematic analysis of the effect of different technical improvements (hierarchical BC, larger Transformer models, larger datasets, online fine-tuning with RL), notably finding that scaling training sets or model size alone will not bridge the gap to the best symbolic agent.

The problem is of very high interest to the AI community, and the technical investigation, results, and discussion appear thorough and insightful. The dataset might also enable further research. I find especially the results regarding scaling interesting, i.e., that performance increases logarithmic, and so more data or bigger models alone will not enable achieving parity with the symbolic approach.

Quality of writing is very good, and so the paper is easy to follow (subject to my lack of technical background).

Minor notes:
 - The paper appears to be missing a link to the dataset
 - The related work is not easy to access for someone not close to the field. E.g., paragraphs on "imitation learning" and "hierarchical policy learning" give too little detail about the basic ideas (do not start with descriptions of what they are for, but what they do)
 - "The full observation space of NLE is far richer and more informed than the view afforded to human players of NetHack, who observe only the more ambiguous “text-based” components of NLE observations" - I do not fully understand this sentence, please expand. What can systems observe in NLE, that humans don't receive in the original interface? Or do you mean that NLE aggregates the Ascii terminal characters into something more high-level?
 - Showing an excerpt from the dataset would be helpful, especially, as it is not quite clear what is added there, both strategies and substrategies? Or the more specific one only?


**Strengths:**

See above.

**Weaknesses:**

See above.

**Questions:**

See above.

**Limitations:**

Yes, the authors critically discuss that scaling alone will not bridge the gap to symbolic agents on this challenge.

---

> ### Author Rebuttal · Authors · 2023-08-10
>
> Dear Reviewer,
>
>
> We are very grateful for the time you have taken to provide thoughtful feedback and comments on our paper.
>
>
> > The paper appears to be missing a link to the dataset.
>
>
> We will publicly release the HiHack dataset upon revision of our submission, at which time we will include a link to the dataset in the main body of the paper.
>
>
> > The related work is not easy to access for someone not close to the field. E.g., paragraphs on "imitation learning" and "hierarchical policy learning" give too little detail about the basic ideas (do not start with descriptions of what they are for, but what they do)
>
>
> Thank you for these suggestions, we will increase the level of detail employed in the related work section of the final version of the paper.
>
>
> > "The full observation space of NLE is far richer and more informed than the view afforded to human players of NetHack, who observe only the more ambiguous “text-based” components of NLE observations" - I do not fully understand this sentence, please expand.
>
>
> The full observation space of NLE includes not only the “text-based” view of NetHack (i.e. the tty characters observed by human players of the game), but also *glyphs* representing the exact identities of all monsters and objects currently in view as well as all items in the player’s inventory [1]. Glyphs are not (directly) observable to human players of the game in standard versions of NetHack. Our HiHack dataset follows the convention set by Hambro et al. [2], storing the text-based, *tty*-view of NetHack observations only.
>
>
> > Showing an excerpt from the dataset would be helpful, especially, as it is not quite clear what is added there, both strategies and substrategies? Or the more specific one only?
>
>
> In its current form, the HiHack dataset reflects high level strategy information from AutoAscend only, as detailed in sections B and C of the supplementary materials accompanying the paper. All hierarchical policies in the paper are trained against these high level strategy labels only. We will add excerpts previewing the dataset to the supplementary materials during revision.
>
>
> Given that there is already interest, we will release an additional version of the dataset featuring both high-level and sub-strategy labels upon revision, reflecting the full structure of AutoAscend displayed in Figure 7 of the supplement.
>
>
> Thank you once again for your review. Please do let us know if you have any outstanding concerns or questions about our work.
>
>
> **Citations**
>
>
> [1] Küttler, Heinrich, et al. "The nethack learning environment." Advances in Neural Information Processing Systems 33 (2020): 7671-7684.
>
>
> [2] Hambro, Eric, et al. "Dungeons and Data: A Large-Scale NetHack Dataset." Advances in Neural Information Processing Systems 35 (2022): 24864-24878.

---

### Official Review · Reviewer_eWjQ · 2023-07-18

**Soundness:** 3 good
**Presentation:** 3 good
**Contribution:** 2 fair
**Rating:** 6
**Confidence:** 2

**Summary:**

The paper explores reasons for this performance gap between neural and symbolic methods in NetHack:
Symbolic agents use hierarchical policies and parsers to extract high-level features
Symbolic agents have handcrafted heuristics and error correction
Neural agents lack inductive biases like hierarchy that may be needed for sparse rewards
Experiments show hierarchy, scale, and combining imitation and RL help improve neural agents:
Hierarchical behavior cloning improves over flat BC
Larger Transformer-based architectures improve over LSTMs
RL fine-tuning provides gains, especially for underfitting models
But significant gaps to symbolic agents remain

**Strengths:**

The experimental design is very clever, the chart is very clear, and the experimental effect is obvious. The paper explores a novel problem domain of applying neural networks to master the game NetHack, where current methods struggle compared to symbolic AI. The authors introduce a new large-scale dataset of NetHack demonstrations called HiHack to facilitate this analysis. The idea of using demonstrations to help neural networks learn better policies in sparse, long-horizon environments like NetHack is creative.The methods are detailed appropriately to replicate experiments. Results are presented logically and incorporate useful visualizations. The conclusion summarizes takeaways concisely.Mastering complex environments like NetHack with sparse rewards and long time horizons remains an open challenge for deep RL. This paper provides significant evidence and analysis characterizing the limitations of current neural network methods in these settings, and points the way towards progress, whether via incorporating stronger inductive biases like hierarchy or combining neural and symbolic approaches. The insights will broadly impact research in sparse reward RL, imitation learning, and integrating neural and classical AI.

**Weaknesses:**

This model is based on the nethack, and the results hold up on the above models, and whether the above results can still hold up on the other models。The authors recognize the limited generality so far of methods tested on NetHack to other complex environments.No obvious harmful biases or problematic data sources are introduced in this work. The NetHack environment itself seems relatively innocuous.


**Questions:**

Can you add some experiments, add some theoretical derivation, whether the contribution of this article is more.

**Limitations:**

The model is not so representative, can switch a more popular model。Overall, the authors demonstrate good care and thoughtfulness regarding the limitations and potential negative impacts of this research direction. The discussion seems sufficient without being overreaching or distracting from the primary technical contributions. I do not have any major suggestions for improvement.

---

> ### Author Rebuttal · Authors · 2023-08-09
>
> Dear Reviewer,
>
>
> We thank you for your feedback and comments on our submission. We are glad that you found our paper to be clear and insightful.
>
>
> > The model is not so representative, can switch a more popular model
>
>
> The transformer-LSTM policy architecture that we employ in this paper is indeed novel, and was developed by us as a result of the limited generalization and expensive gradient updates that we observed when experimenting with more standard transformer-only architectures.
>
>
> A long context length appears to be greatly beneficial to successful imitation of AutoAscend. We found augmentation of a causal transformer-core with an LSTM recurrent module to offer a simple and, importantly, highly lightweight solution to this issue.
>
>
> Please take a look at **section (ii)** of our general rebuttal above as well as the attached PDF for additional reasoning and figures supporting these claims.
>
>
> > This model is based on the nethack, and the results hold up on the above models, and whether the above results can still hold up on the other models
>
>
> As above, please refer to **section (i)** of our general rebuttal for our response to this point. We will add this additional discussion to the paper.
>
>
> Thank you once again for the time you have taken to review our paper, and please let us know if you have any outstanding concerns that stand between us and a strong recommendation for acceptance.

---

### Author Rebuttal · Authors · 2023-08-09

We thank the reviewers for their constructive and insightful feedback. We are glad that you found our analysis to be comprehensive (qyRc), our experimental insights impactful (eWjQ, Ub8t, 83YB), and our submission to be well-written (Ub8t, qyRc, 83YB).

However, in this general response we would like to address two concerns raised by several reviewers: the applicability of this work outside of the NetHack Learning Environment (eWjQ, PtAe, qyRc) and transformer-only architectures (eWjQ, PtAe).

**(i) On the limited applicability of this work outside of NLE**

While we understand the sentiment behind this statement, we respectfully disagree that our experimental insights are limited to NLE. Performance gaps between neural and symbolic methods are ubiquitous in open-ended environments including task and motion planning (TAMP) settings, and their underlying causes have received little study [1, 2].  Our study fills this gap in the neural policy learning literature by conducting a scientific investigation of the failure modes of popular neural approaches at mastering complex, generalizable behaviors directly from multi-modal data.


NLE is well-suited for such an investigation on account of:
(1) the sheer extent of the performance gap between leading symbolic and neural policies [3, 4];
(2) the existence of previously-conducted benchmarks [3, 4]; and
(3) the open-source and hierarchical nature of the symbolic, state-of-the-art agent, AutoAscend [3].


In particular, we believe the last point makes NetHack a singularly compelling testbed over peer environments such as Habitat, MineCraft, or AI2-Thor. In generating the large-scale HiHack dataset, we provide the community with what is, to our knowledge, a unique opportunity to explore the impact of hard-coded hierarchical behavioral priors via structured labels on learning in long-horizon, complex environments.


Further, we take measures to preserve the generality of the insights yielded by our investigations of neural policy learning in this paper. Specifically, we intentionally forgo the addition of any environment-specific constraints in the architectural design or training setup of all models explored in this paper. This contrasts with leading NLE agents RAPH and KakaoBrain that rely on augmentations such as hand-engineered separated action spaces, role-specific training, and hard-coded sub-routines [3]. While this choice prevents us from achieving absolute SOTA in NLE in this paper, we believe it to be crucial in preserving the general applicability of our insights to neural policy learning for general open-ended, long-horizon environments.


We agree with the reviewers that further probing of the nature of tradeoffs between hard-coded hierarchical priors, model capacity, and interaction in other environments is an important and exciting direction for future work, but we believe it to be beyond the scope of this paper.


**(ii) On transformer-only architectures**

The first iteration of transformer-based models we experimented with in our investigations were structured precisely as several of the reviewers described, featuring a “flat” transformer core module only (i.e. with the LSTM-based recurrent module ablated from the model architecture visualized in Figure 4 (left)), meaning all input is now confined to the transformer’s context window. We found such models to perform substantially worse when trained with behavioral cloning on AutoAscend demonstration data than their pure LSTM or transformer-LSTM counterparts.


In the PDF attachment, please find visualizations of the flat transformer policies’ behavioral cloning rolling score [4] curves on withheld seeds of NLE over the course of training, contrasted against the corresponding curves for the transformer + LSTM and hierarchical transformer + LSTM policies (included in Figure 13 of our submission’s supplement).


The parameter “unroll length,” denoted as “URL” in the legend of this figure, reflects the length of the context or observation history employed for action prediction.





A total of six random seeds for each model class were employed to test flat transformer policies. Training for each random seed was conducted on a single GPU for 48 hours. The transformer core modules of these policies have precisely the same architectural hyperparameters as their counterparts in the transformer-LSTMs, but featuring 6 layers.


We make two key observations: (1) the number of samples seen during the 48-hour training window is inversely proportional to the context length employed in training; (2) the rate of BC policy improvement does not increase with context length, across the set of context length values tested here. It is on this basis of (2), coupled with the fact that the average AutoAscend demonstration in our HiHack dataset consists of 27000 keypresses (Table 1 in the paper) – several orders of magnitude beyond the longest context length, URL = 256, we find feasible to test for flat transformers – that we conclude that an encoding of full game history is important for imitation in our setting.


We will update the supplemental section of our paper to include an in-depth discussion of these flat transformer ablation experiments upon revision.




**Citations**


[1] Silver, Tom, et al. "Learning symbolic operators for task and motion planning." 2021 IEEE/RSJ International Conference on Intelligent Robots and Systems (IROS). IEEE, 2021.


[2] Zhang, Kai, et al. "Task and motion planning methods: applications and limitations." 19th International Conference on Informatics in Control, Automation and Robotics ICINCO 2022). SCITEPRESS-Science and Technology Publications, 2022.


[3] Hambro, Eric, et al. "Insights from the neurips 2021 nethack challenge." NeurIPS 2021 Competitions and Demonstrations Track. PMLR, 2022.


[4] Hambro, Eric, et al. "Dungeons and Data: A Large-Scale NetHack Dataset." Advances in Neural Information Processing Systems 35 (2022): 24864-24878.

---

### Decision · Program_Chairs · 2023-09-21

**Decision:**

Accept (poster)

**Comment:**

After careful consideration, I recommend that the submission be accepted to NeurIPS 2023. The reviewers agree that the paper tackles an important problem and presents a novel and clever solution with thorough and insightful experimentation. I recommend the authors to include the changes suggested by the reviewers in the camera-ready version including experimentation with vanilla transformer and open-source code for the proposed method.